# Revisiting the Sample Complexity of Sparse Spectrum Approximation of Gaussian Processes

**Quang Minh Hoang**
Department of Computer Science
Carnegie-Mellon University
Pittsburgh, PA 15213
qhoang@andrew.cmu.edu

**Trong Nghia Hoang**
MIT-IBM Watson AI Lab
IBM Research
Cambridge, MA 02142
nghiaht@ibm.com

**Hai Pham**
Language Technologies Institute
Carnegie-Mellon University
Pittsburgh, PA 15213
htpham@cs.cmu.edu

**David P. Woodruff**
Department of Computer Science
Carnegie-Mellon University
Pittsburgh, PA 15213
dwoodruf@cs.cmu.edu

## Abstract

We introduce a new scalable approximation for Gaussian processes with provable guarantees which hold simultaneously over its entire parameter space. Our approximation is obtained from an improved sample complexity analysis for sparse spectrum Gaussian processes (SSGPs). In particular, our analysis shows that under a certain data disentangling condition, an SSGP's prediction and model evidence (for training) can well-approximate those of a full GP with low sample complexity. We also develop a new auto-encoding algorithm that finds a latent space to disentangle latent input coordinates into well-separated clusters, which is amenable to our sample complexity analysis. We validate our proposed method on several benchmarks with promising results supporting our theoretical analysis.

## 1 Introduction

A Gaussian process (GP) [36] is a popular probabilistic kernel method for regression which has found applications across many scientific disciplines. Examples of such applications include meteorological forecasting, such as precipitation and sea-level pressure prediction [2]; sensing and monitoring of ocean and freshwater phenomena such as temperature and plankton bloom [7, 12]; traffic flow and mobility demand predictions over urban road networks [9, 10, 29]; flight delay predictions [15, 19, 20]; and persistent robotics tasks such as localization and filtering [43]. The broad applicability of GPs is in part due to their expressive Bayesian non-parametric nature which provides a closed-form prediction [36] in the form of a Gaussian distribution with formal measures of predictive uncertainty, such as entropy and mutual information criteria [27, 39, 44]. Such expressiveness makes GPs not only useful as predictive methods but also a go-to representation for active learning applications [24, 23, 27, 44] or Bayesian optimization [38, 45, 22, 16] that need to optimize for information gain while collecting training data.

Unfortunately, the expressive power of a GP comes at a cost of poor scalability (i.e., cubic time [36]) in the size of the training data (see Section 2.1 below), hence limiting its use to small datasets. This prevents GPs from being applied more broadly to modern settings with increasingly growing volumes of data [15, 19, 20]. To sidestep this limitation, a prevalent research trend is to impose sparse structural assumptions [33, 34] on the GP's kernel matrix to reduce its multiplication and inversion cost, which comprises the main bulk of the training and inference complexity. This results in a broad family of

sparse Gaussian processes [15, 17, 19, 28, 37, 40] that are not only computationally efficient but also amenable to various forms of parallelism [8, 29] and distributed computation [1, 13, 20, 21, 18], further increasing their efficiency.

Despite such advantages, the sparsification components at the core of these methods are heuristically designed and do not come with provable guarantees that explicitly characterize the interplay between approximation quality and computational complexity. This motivates us to develop a more robust, theoretically-grounded approximation scheme for GPs that is both provable and amenable to the many fast computation schemes mentioned above. More specifically, our contributions include:

**1.** An analysis of a new approximation scheme that generates a sparse spectrum approximation of a GP with provable bounds on its sample complexity, which practically becomes significantly small when the input data exhibits a certain clustering structure. Furthermore, the impact of the approximation on the resulting training and inference qualities is also formally analyzed (Section 3.1).

**2.** A data partitioning algorithm inspired from the above analysis, which learns a cluster embedding that reorients the input distribution while ensuring reconstructability of the original distribution (Section 3.3). We show that using sparse spectrum Gaussian processes (SSGP) [17, 28] on the embedded space requires fewer samples to achieve the same level of approximation quality. This also induces a linear feature map which enables efficient training and inference of GPs.

**3.** An empirical study on benchmarks that demonstrates the efficiency of the proposed method over existing works in terms of its approximation quality versus computational efficiency (Section 4).

## 2 Related Work

In this section we provide an overview of Gaussian processes (Section 2.1), followed by a succinct summary of their spectral representations (Section 2.2).

### 2.1 Gaussian Processes (GPs)

A Gaussian process [36] defines a probabilistic prior over a random function $g(\mathbf{x})$ defined by mean function $m(\mathbf{x}) = 0^1$ and kernel function $k(\mathbf{x}, \mathbf{x}')$. These functions induce a marginal Gaussian prior over the evaluations $\mathbf{g} = [g(\mathbf{x}_1) \dots g(\mathbf{x}_n)]^\top$ on an arbitrary finite subset of inputs $\{\mathbf{x}_1, \dots, \mathbf{x}_n\}$. Let $\mathbf{x}_*$ be an unseen input whose corresponding output $g_* = g(\mathbf{x}_*)$ we wish to predict. The Gaussian prior over $[g(\mathbf{x}_1) \dots g(\mathbf{x}_n) \, g(\mathbf{x}_*)]^\top$ implies the following conditional distribution:

$$g_* \triangleq g(\mathbf{x}_*) \mid \mathbf{g} \quad \sim \quad \mathbf{N}\left(\mathbf{k}_*^\top \mathbf{K}^{-1}\mathbf{g}, \; k(\mathbf{x}_*, \mathbf{x}_*) - \mathbf{k}_*^\top \mathbf{K}^{-1}\mathbf{k}_*\right), \tag{1}$$

where $\mathbf{k}_* = [k(\mathbf{x}_*, \mathbf{x}_1) \dots k(\mathbf{x}_*, \mathbf{x}_n)]^\top$ and $\mathbf{K}$ denotes the Gram matrix induced by $k(\mathbf{x}, \mathbf{x}')$ on $\{\mathbf{x}_1, \dots, \mathbf{x}_n\}$ for which $\mathbf{K}_{ij} = k(\mathbf{x}_i, \mathbf{x}_j)$. For a noisy observation $y$ perturbed by Gaussian noise such that $y \sim \mathbf{N}(g(\mathbf{x}), \sigma^2)$, Eq. (1) above can be integrated with $\mathbf{N}(\mathbf{g}, \sigma^2\mathbf{I})$ to yield:

$$g_* \triangleq g(\mathbf{x}_*) \mid \mathbf{y} \quad \sim \quad \mathbf{N}\left(\mathbf{k}_*^\top (\mathbf{K} + \sigma^2\mathbf{I})^{-1}\mathbf{y}, \; k(\mathbf{x}_*, \mathbf{x}_*) - \mathbf{k}_*^\top (\mathbf{K} + \sigma^2\mathbf{I})^{-1}\mathbf{k}_*\right), \tag{2}$$

which explicitly forms the predictive distribution of a Gaussian process. The defining parameter $\boldsymbol{\Theta}$ of $k(\mathbf{x}, \mathbf{x}')$ (see Section 2.2) is crucial to the predictive performance and needs to be optimized via minimizing the negative log likelihood of $\mathbf{y}$:

$$\ell(\boldsymbol{\Theta}) \quad = \quad \frac{1}{2}\log\left|\mathbf{K}_{\boldsymbol{\Theta}} + \sigma^2\mathbf{I}\right| + \frac{1}{2}\mathbf{y}^\top\left(\mathbf{K}_{\boldsymbol{\Theta}} + \sigma^2\mathbf{I}\right)^{-1}\mathbf{y}, \tag{3}$$

where we now use the subscript $\boldsymbol{\Theta}$ to indicate that $\mathbf{K}$ is a function of $\boldsymbol{\Theta}$. In practice, both training $\boldsymbol{\Theta}$ and prediction incur $\mathbf{O}(n^3)$ processing cost, which prevents direct use of Gaussian processes on large datasets that might contain more than tens of thousands of training inputs.

### 2.2 Sparse Spectrum Gaussian Processes

Sparse spectrum Gaussian processes (SSGPs) [14, 17, 28] exploit Theorem 1 below to re-express the Gaussian kernel $k(\mathbf{x}, \mathbf{x}') \triangleq \exp(-0.5\,(\mathbf{x} - \mathbf{x}')^\top \boldsymbol{\Theta}^{-1}\,(\mathbf{x} - \mathbf{x}'))$ (where $\boldsymbol{\Theta} \triangleq \mathrm{diag}[\theta_1^2 \dots \theta_d^2]$) as an integration over a spectrum of cosine functions such that the integrating distribution (over the frequencies that parameterize these functions) is a multivariate Gaussian.

**Theorem 1** (Bochner Theorem). *Let $k(\mathbf{x}, \mathbf{x}')$ denote a Gaussian kernel defined above and let $p(\mathbf{r}) \sim \mathbf{N}(\mathbf{0}, (4\pi^2\mathbf{\Theta})^{-1})$. It follows that:*

$$k(\mathbf{x}, \mathbf{x}') = \mathbb{E}_{\mathbf{r} \sim p(\mathbf{r})}\left[\cos\left(2\pi\mathbf{r}^\top(\mathbf{x} - \mathbf{x}')\right)\right], \tag{4}$$

*where $\mathbf{r}$ is a $d$-dimensional random variable that parameterizes $\cos(2\pi\mathbf{r}^\top(\mathbf{x} - \mathbf{x}'))$. In practice, $\mathbf{r}$ is often referred to as the spectral frequency.*

This allows us to approximate the original Gram matrix $\mathbf{K}$ with a low-rank matrix $\mathbf{K}'$ constructed by a linear kernel $\mathbf{K}'(\mathbf{x}, \mathbf{x}') = \mathbf{\Phi}(\mathbf{x})^\top\mathbf{\Phi}(\mathbf{x})$ with feature map $\mathbf{\Phi}(\mathbf{x}) = [\phi_1(\mathbf{x}) \dots \phi_{2m}(\mathbf{x})]^\top$ comprising $2m$ basis trigonometric functions [17]. Each pair of odd- and even-index basis functions $\phi_{2i-1}(\mathbf{x}) = \cos(2\pi\mathbf{r}_i^\top\mathbf{x})$ and $\phi_{2i}(\mathbf{x}) = \sin(2\pi\mathbf{r}_i^\top\mathbf{x})$ is parameterized by the same sample of spectral parameter $\mathbf{r}_i \sim p(\mathbf{r})$. For efficient computation, $m$ is often selected to be significantly smaller than $n$ (i.e., the number of training examples). However, to guarantee that $\|\mathbf{K} - \mathbf{K}'\|_2 \leq \lambda$ with probability at least $1 - \delta$, $m$ needs to be as large as $\mathbf{O}(n^2/\lambda^2 \log(n/\delta))$ [31][2], which makes the total prediction complexity much worse than that of a full GP.

Alternatively, one can use kernel sketching methods [3, 32, 35] to generate feature maps that scale more favorably with the effective dimension of the kernel matrix, which empirically tends to be on the order of $\mathbf{O}(\log n)$. However, the pitfall of these methods is that without knowing the exact parameter configuration $\mathbf{\Theta}$ that underlies the data, they cannot sample from the true $p(\mathbf{r})$, which is necessary in their analyses. As such, existing random maps [3, 35] that were generated based on this spectral construction often depend on a parameter initialization and their approximation quality is only guaranteed for that particular parameter setting instead of uniformly over the entire parameter space. This motivates us to revisit the sample complexity of SSGP from a setting which specifically searches for a reorientation of the input distribution such that the reoriented data exhibits a disentangled cluster structure. Such disentanglement provides a more sample-efficient bound as we show in our analysis in Section 3.1 below.

# 3 Provable Approximation of SSGPs with Improved Sample Complexity

We first show how a sparse spectrum Gaussian process (SSGP) [28] can be approximated well with a provably low sample complexity. This is achieved by revisiting its sample complexity which, unlike prior work [3, 31, 35], explicitly characterizes and accounts for a certain set of data disentanglement conditions. Importantly, our new analysis (Section 3.1) yields practical bounds on both an SSGP's prediction and model evidence (Section 3.2) that hold with high probability uniformly over the entire parameter space[3]. Furthermore, our analysis also inspires an encoding algorithm that finds a latent space to disentangle the encoded coordinates of data into well-separated clusters on which a sparse spectrum GP can approximate a GP provably well (Section 3.3). Our experiments show that such a latent space can be found for several real-world datasets (Section 4).

## 3.1 Practically Improved Sample Complexity for Sparse Spectrum Gaussian Processes

This section derives a new data-oriented feature map to approximate a Gaussian process parameterized with a Gaussian kernel. Unlike existing work which assumes knowledge of the true kernel parameters [3, 32, 35], our derivation remains oblivious to such parameters, and therefore holds universally over their entire candidate space. We assume that the GP prior of interest is of the form $g(\mathbf{x}) \sim \mathrm{GP}(0, k(\mathbf{x}, \mathbf{x}'))$ where $k(\mathbf{x}, \mathbf{x}')$ represents its Gaussian kernel in Section 2.2.

We give our analysis in three parts: (1) the spectral sampling scheme and a notion of approximation loss; (2) a set of practical data conditions which can be either observed from a raw data distribution or approximately imposed on the data via a certain embedding; (3) a theoretical analysis that delivers our key result that establishes an improved sample complexity when our data conditions are met.

### 3.1.1 Spectral Sampling Scheme and Spectral Loss

We show that $g(\mathbf{x})$ can be approximated by $g'(\mathbf{x}) = \sum_{i=1}^{p} g_i(\mathbf{x})$ with provable data-oriented guarantees where $g_i(\mathbf{x}) \sim \mathrm{GP}(0, (1/\sqrt{p})k_i(\mathbf{x}, \mathbf{x}'))$. To achieve this, we first establish in Lemma 1

that the induced Gram matrix $\mathbf{K}$ of $k(\mathbf{x}, \mathbf{x}')$ on any dataset can be represented as an expectation over a space of induced Gram matrices $\{\mathbf{K}_i\}_{i=1}^p$ produced by a corresponding space of random kernels $\{k_i(\mathbf{x}, \mathbf{x}')\}_{i=1}^p$.

**Lemma 1.** *Let $k(\mathbf{x}, \mathbf{x}')$ and $\mathbf{K}$ denote a Gaussian kernel parameterized by $\boldsymbol{\Theta}$ (Section 2.2) and its induced Gram matrix on an arbitrary set of training inputs, respectively. There exists a space $\mathcal{K}$ of random kernels $\kappa(\mathbf{x}, \mathbf{x}')$ and a $\boldsymbol{\Theta}$-independent distribution $\rho$ over $\mathcal{K}$ for which $\mathbf{K} = \mathbb{E}_\kappa[\mathbf{K}_\kappa]$ where $\mathbf{K}_\kappa$ denotes the induced Gram matrix of $\kappa$ on the same set of training inputs.*

This follows directly from Theorem 1 above which states that $k(\mathbf{x}, \mathbf{x}') = \mathbb{E}[\cos(2\pi\mathbf{r}^\top(\mathbf{x}-\mathbf{x}'))]$ where $\mathbf{r} \sim \mathbf{N}(\mathbf{0}, (4\pi^2\boldsymbol{\Theta})^{-1})$. We can choose $\kappa(\mathbf{x}, \mathbf{x}'; \boldsymbol{\epsilon}) = \cos(\boldsymbol{\epsilon}^\top\boldsymbol{\Theta}^{-0.5}(\mathbf{x} - \mathbf{x}'))$ where $\boldsymbol{\epsilon} \sim \mathbf{N}(\mathbf{0}, \mathbf{I})$ which implies $\mathbf{k}(\mathbf{x}, \mathbf{x}') = \mathbb{E}_\epsilon[\kappa(\mathbf{x}, \mathbf{x}'; \boldsymbol{\epsilon})]$. Thus, $\mathbf{K} = \mathbf{E}_\epsilon[\mathbf{K}_\epsilon]$ where the $\boldsymbol{\Theta}$-independent parameter $\boldsymbol{\epsilon}$ indexes $\kappa$ and $\mathbf{K}_\epsilon$ is the induced Gram matrix of $\kappa$. Leveraging the result of Lemma 1, a naïve analysis [31] using worst-case concentration bounds to derive a conservative estimate for a sufficient number of samples would require a prohibitively expensive sample complexity of $\mathbf{O}(n^2 \log n)$.

Such analyses, however, often ignore the input distribution, which can be used to sample more selectively, thereby significantly reducing the sample complexity. This is demonstrated below in Theorem 2 which shows that when the input distribution exhibits a certain degree of compactness and separation (as defined in Conditions 1-3), we only require $\mathbf{O}((\log^2 n/\lambda^2) \log \log(n/\delta))$ sampled kernels $\{\kappa_i\}_{i=1}^p$ indexed by $\{\boldsymbol{\epsilon}_i\}_{i=1}^p$ to produce an average Gram matrix $\mathbf{K}' = \frac{1}{p}\sum_{i=1}^p \mathbf{K}_{\epsilon_i}$ that is sufficiently close to $\mathbf{K}$ in spectral norm (see Definition 1) with probability at least $1 - \delta$.

**Definition 1** (Spectral Closeness). *Given $\lambda > 0$, the symmetric matrices $\mathbf{K}$ and $\mathbf{K}'$ are $\lambda$-close if $\|\mathbf{K} - \mathbf{K}'\|_2 \leq \lambda$ where $\|\mathbf{K} - \mathbf{K}'\|_2 = \lambda_{\max}(\mathbf{K} - \mathbf{K}')$ denotes the largest eigenvalue of $\mathbf{K} - \mathbf{K}'$.*

Thus, parameterizing the GP prior with $\mathbf{K}'$ instead of $\mathbf{K}$ allows us to derive an upper bound on the expected difference between their induced model evidence (for learning kernel parameters) and prediction losses (for testing) with respect to the same parameter setup (Theorem 3). Theorem 3 importantly exploits the fact that the bound in Theorem 2 holds universally over the entire space of parameters, which allows us to bound the prediction difference between the original and approximated GPs with respect to their own optimized parameters (that are not necessarily the same).

### 3.1.2 Practical Conditions on Data Distributions

We now outline key practical data conditions, which can be satisfied approximately via an encoding algorithm that transforms the input data into a latent space where such conditions are met. These conditions are necessary for deriving a practically improved sample complexity in Section 3.1.3.

**Condition 1.** For each parameter configuration $\boldsymbol{\Theta} = \mathrm{diag}[\theta_1^2, \ldots, \theta_d^2]$, there exists a mixture distribution $\mathcal{M}(\mathbf{x}; \boldsymbol{\gamma} = (\gamma_1, \ldots, \gamma_b), \boldsymbol{\pi} = (\pi_1, \ldots, \pi_b), \mathbf{c} = (\mathbf{c}_1, \ldots, \mathbf{c}_b))$ with at most $b = \mathbf{O}(\log n)$ Gaussian components $\mathbf{N}(\mathbf{x}; \mathbf{c}_i, \gamma_i^2\boldsymbol{\Theta}^{-1})$ over the data space with the mixing weights $\pi_i \propto 2^{\frac{i}{2}}$ and variances $\gamma_i = \mathbf{O}(\frac{1}{\sqrt{d}})$ that *generate* the observed data in $d$-dimensional space.

**Condition 2**. The $i^{\text{th}}$ Gaussian component as defined in Condition **1** above was used to generate $2^{\frac{i}{2}}$ data points of the observed dataset. This can be substantiated easily with high probability given the above setup in Condition **1** that assigns selection probability $\pi_i \propto 2^{\frac{i}{2}}$ to the $i^{\text{th}}$-component.

**Condition 3**. For each parameter configuration $\boldsymbol{\Theta} = \mathrm{diag}[\theta_1^2, \ldots, \theta_d^2]$, the mixture distribution of data in Condition **1** has sufficiently separated cluster centers. That is, for all $i \neq j$:

$$\left\|\boldsymbol{\Theta}^{-1/2}(\mathbf{c}_i - \mathbf{c}_j)\right\|_2^2 \; > \; \frac{3}{2}\log\left(\frac{2^a}{2^a - 1}\right) \qquad \text{where} \qquad a \; = \; \frac{1}{\log 2}\log\left(\frac{n^4}{n^4 - \lambda^4}\right). \qquad (5)$$

These conditions impose that the observed data can be separated into a number of clusters with exponentially growing sizes and concentration (see the small variances defined in Condition **1** and the imposed sizes of Condition **2**). Intuitively, this means data points that belong to clusters with high concentration are responsible for kernel entries with high values whereas those in clusters with low concentration generate entries with low values. This is easy to see since high concentration reduces the distance between data points, thus increasing their kernel values and vice versa.

Furthermore, as imposed by Condition **2**, clusters with high concentration also have denser population and induce kernel entries with high value. In addition, Condition **3** requires that clusters are well-separated, which implies that a large number of kernel entries are small and therefore can be

approximated cheaply. Together, these conditions form the foundations of our reduced complexity analysis for SSGP in Theorem 2. Interestingly, we show that such conditions also inspire the development of a probabilistic algorithm that finds an encoding of the input that (approximately) satisfies these conditions while preserving the statistical properties of the input (Section 3.3). This results in an improved sample complexity for SSGPs in practice (see Section 3.1.3).

### 3.1.3 Main Results

To understand the intuition why an improved sample complexity can be obtained, we note that when data is partitioned in clusters with different concentrations and sizes, the kernel entries are also partitioned into multiple value-bands with narrow width (i.e., low variance). Exploiting this, we can calibrate a significantly lower sample complexity for each band using concentration inequalities that improve with lower variance [11, 25].

Then, to combine these in-band sample complexities efficiently, we further exploit the data conditions in Section 3.1.2 to show that statistically, value bands with smaller width also tend to be populated more densely[4]. This allows us to aggregate these in-band sample costs into an overall sample complexity with low cost. In practice, this also inspires an embedding algorithm (Section 3.3) that transforms the data in such a way that the distribution of their induced kernel entries will be denser in narrower bands, which is advantageous in our analysis.

Formally, let $\mathcal{C}$ be the set of all kernel entries indexed by $(u, v)$ in the Gram matrix $\mathbf{K}$ such that $\mathbf{x}_u$ and $\mathbf{x}_v$ belong to the same cluster and $\mathcal{C}'$ be its complement. Also, let $\mathcal{C}$ be partitioned[5] into $b$ value-bands $\kappa_i = \{(u, v) \in \mathcal{C} \mid 1 - \mathbf{O}(2^{1-i}) \leq \mathbf{K}_{uv}^4 \leq 1 - \mathbf{O}(2^{-i})\}$ for $i \in [1 \dots b]$ and let $\kappa_0 = \{(u, v) \in \mathcal{C} \mid \mathbf{K}_{uv}^4 \geq 1 - \mathbf{O}(2^{-b})\}$ be a band that is only populated by very large kernel entries. Theorem 2 below shows that we can construct a $\lambda$-spectral approximation of $\mathbf{K}$ with arbitrarily high probability and low sample complexity.

**Theorem 2.** *For any $1 \geq \delta \geq \mathbf{O}(\exp(b - \sqrt{d}))$, if the training data has $n$ data points and satisfies Conditions **1-3** above with respect to $\lambda$, then with probability at least $1 - 2\delta$, the approximation $\mathbf{K}' = (1/p) \sum_{i=1}^{p} \mathbf{K}_{\epsilon_i}$ where $\epsilon_i \sim \mathbf{N}(\mathbf{0}, \mathbf{I})$ is $\lambda$- spectral close to $\mathbf{K}$.*

**Proof Sketch.** Our proof strategy is outlined below. The formal statements are spelled out in Appendix A.

First, with a proper choice of a clustering partition, the cross-cluster entries in $\mathbf{K}$ are guaranteed to be sufficiently small so as to be well-approximated by zero. We can then show with high probability that any kernel entry that corresponds to a pair of unique data points from the same cluster can be well-approximated with a sample complexity that scales favorably with the cluster's variance. In particular, we show that kernel values induced by data points generated by lower-variance clusters (see Condition **1**) will have smaller approximation variances than those generated by data from higher-variance clusters and therefore require fewer samples to produce the same level of approximation.

Second, for certain configurations of mixture weights, Condition **2** asserts that the number of data points from each cluster is inversely proportional to the cluster variance, which implies that a small sample complexity is enough to approximate the majority of kernel entries. More specifically, Lemma 3 shows that when the input points are distributed into clusters with certain choices of variances $\{\gamma_i\}_{i=1}^{b}$ and at an inversely proportional ratio $\mathbf{O}(\gamma_i^{-1})$, then with high probability, over all clusters, the kernel entries (excluding those on the diagonal) associated with pairs in the $i$-th cluster belong to their corresponding band $\kappa_i$.

Lemma 5 shows that for $p = \mathbf{O}(\log^2 n/\lambda^2 \log(\log n/\delta))$, with probability $1 - \delta/b$, the total approximation error of all kernel entries in the $\mathcal{C}_i$ will be at most $\lambda^2/4b$, which implies with probability $1 - \delta$, the total approximation cost for items in $\mathcal{C}$ is at most $\lambda^2/4$. Next, Lemma 2 establishes that with the above data distribution, $\mathcal{C}$ accounts for $n^2/4$ entries while $\mathcal{C}'$ accounts for $3n^2/4$ entries, which needs to be approximated with error at most $3\lambda^2/4$.

Finally, Lemma 4 shows that when the clusters are sufficiently well-separated (see Condition **3**), any kernel value corresponding to an arbitrary data pair with points belonging to different clusters is guaranteed to be smaller than $\lambda^2/n^2$, which then guarantees a total error of at most $3\lambda^2/4$ when they are uniformly approximated with zero. Putting these together yields a total error of $\lambda^2$ with probability

$1 - 2\delta$, which implies $\mathbf{K}$ and $\mathbf{K}'$ are $\lambda$-spectrally close since $\|\mathbf{K} - \mathbf{K}'\|_2 \leq \|\mathbf{K} - \mathbf{K}'\|_{\mathrm{F}} \leq \lambda$. Please see Appendix A for details.

## 3.2 Approximation Loss for Prediction and Model Evidence

In terms of prediction and model evidence approximation, our result holds simultaneously for all parameter configurations and is thus oblivious to the choice of parameters (see Theorem 3). While existing kernel sketch methods [3, 32] generically achieve near-linear complexity for the approximate feature map[6], they often require knowledge of the parameters to construct the kernel approximations. In contrast, our result in Theorem 2 can be leveraged to bound the same prediction discrepancy when the original and approximated GPs use their own optimized parameter configurations, as shown in Theorem 4 below. To establish Theorem 4, however, we first establish an intermediate result that bounds the prediction and model evidence in the case when both the original and approximated GPs use the same parameter configurations.

**Theorem 3.** *Let $\delta < 1$ be a user-specified confidence as defined previously in Theorem 2 and let $\mathbf{K}'$ be an approximation to $\mathbf{K}$ for which $\|\mathbf{K} - \mathbf{K}'\|_2^2 \leq \lambda^2$ with probability $1 - \delta$, uniformly over the entire parameter space. Then, with probability $1 - \delta$, the following hold:*

$$\mathbb{E}[g(\mathbf{x}_*)] \;=\; \left(1 \pm \frac{\lambda}{\sigma^2}\right) \mathbb{E}[g'(\mathbf{x}_*)] \qquad and \qquad \mathbb{V}[g(\mathbf{x}_*)] \;=\; \left(1 \pm \frac{\lambda}{\sigma^2}\right) \mathbb{V}[g'(\mathbf{x}_*)] \;\pm\; \frac{\lambda}{\sigma^2} \quad (6)$$

*where $\sigma^2$ is the noise of the variance (Eq. (2)), and $g(\mathbf{x}_*)$, $g'(\mathbf{x}_*)$ respectively denote the predictive distributions of the full GP and the approximated GP pertaining to an arbitrary test input $\mathbf{x}_*$.*

*Proof.* This follows directly from Lemma 7 and Lemma 8 in Appendix B. $\qquad\square$

Finally, Theorem 4 analyzes how close the approximated predictive mean is to the full GP predictive mean when both are evaluated at the optimizer of their respective training objective.

**Theorem 4.** *Let $\delta < 1$ be a user-specified confidence as defined in Theorem 2. Let $\mathbf{K}'$ denote an approximation to $\mathbf{K}$ for which $\|\mathbf{K} - \mathbf{K}'\|_2^2 \leq \lambda^2$ with probability at least $1 - \delta$ uniformly over the entire parameter space. Let $\mathbf{\Theta}_*$ and $\mathbf{\Theta}'_*$ denote the optimal hyperparameters obtained by respectively minimizing the negative log likelihood of a full GP and the approximated GP. With probability $1 - \delta$, the following holds:*

$$\mathbb{E}[g'(\mathbf{x}_*; \mathbf{\Theta}'_*)] \;=\; \left(1 \pm \rho(\lambda, \sigma, \mathbf{\Theta}_*, \mathbf{\Theta}'_*)\right) \cdot \mathbb{E}[g(\mathbf{x}_*; \mathbf{\Theta}_*)] \;+\; \wp(\lambda, \sigma, \mathbf{\Theta}_*, \mathbf{\Theta}'_*) \qquad (7)$$

*where $\rho(\lambda, \sigma, \mathbf{\Theta}_*, \mathbf{\Theta}'_*)$ and $\wp(\lambda, \sigma, \mathbf{\Theta}_*, \mathbf{\Theta}'_*)$ are constants with respect to $\lambda, \sigma, \mathbf{\Theta}_*, \mathbf{\Theta}'_*$.*

*Proof.* This follows immediately from Lemma 11 in Appendix B, which was built on the result of Theorem 3 above. This completes our loss analysis for SSGPs. $\qquad\square$

## 3.3 Optimizing Feature Map Complexity

We next present a practical probabilistic embedding algorithm that transforms the input data to meet the requirements of Conditions **1**-**3**. Our method is built on the rich literature of variational auto-encoders (VAE) [26], which is a broad class of deep generative models that combine the rigor of Bayesian methods and rich parameterization of (deep) neural networks to discover (non-linear) low-dimensional embeddings of data while preserving their statistical properties. We first provide a short review on VAEs below, followed by an augmentation that aims to achieve the impositions in Conditions **1**-**3** above.

### 3.3.1 Variational Auto-Encoders (VAEs)

Let $\mathbf{x}$ be a random variable with density function $p(\mathbf{x})$. We want to learn a latent variable model $p_\theta(\mathbf{x}, \mathbf{z}) = p(\mathbf{z})p_\theta(\mathbf{z}|\mathbf{x})$ that captures this generative process. The latent variable model comprises a fixed latent prior $p(\mathbf{z})$ and a parametric likelihood $p_\theta(\mathbf{z}|\mathbf{x})$. To learn $\theta$, we maximize the variational evidence lower-bound (ELBO) $\mathbf{L}(\mathbf{x}; \theta, \phi)$ of $\log p_\theta(\mathbf{x})$:

$$\mathbf{L}(\mathbf{x}; \theta, \phi) \;\triangleq\; \mathbb{E}_{\mathbf{z} \sim q_\phi}\Big[\log p_\theta(\mathbf{x}|\mathbf{z})\Big] - \mathbb{KL}\Big(q_\phi(\mathbf{z}||\mathbf{x})||p(\mathbf{z})\Big) \qquad (8)$$

with respect to an arbitrary posterior surrogate $q_\phi(\mathbf{z}|\mathbf{x}) \simeq p_\theta(\mathbf{z}|\mathbf{x})$ over the latent variable $\mathbf{z}$. The ELBO is always a lower-bound on $\log p_\theta(\mathbf{x})$ regardless of our choice of $q_\phi(\mathbf{z}|\mathbf{x})$. This is due to the non-negativity of the KL divergence as seen in the first part of the above equation.

This can be viewed as a stochastic auto-encoder with $p_\theta(\mathbf{x}|\mathbf{z})$ and $q_\phi(\mathbf{z}|\mathbf{x})$ acting as the encoder and decoder, respectively. Here, $\theta$ and $\phi$ characterize the neural network parameterization of these models. Their learning is enabled via a re-parameterization of $q_\phi(\mathbf{z}|\mathbf{x})$ that enables stochastic gradient ascent.

### 3.3.2 Re-configuring Data via an Augmenting Variational Auto-Encoder

To augment the above VAE framework [26, 30] to account for the impositions in Conditions **1** and **2**, we ideally want to configure the parameterization of the above generative process to guarantee that the marginal posterior $q(\mathbf{z}) = \int_\mathbf{x} q(\mathbf{z}|\mathbf{x})p(\mathbf{x})\mathrm{d}\mathbf{x}$ will manifest itself in the form of a mixture of Gaussians with the desired concentration and population densities as stated in Condition **1**.

However, it is often difficult to make such an imposition directly given that we typically have no prior knowledge of $p(\mathbf{x})$. We instead impose the desired structure on the latent prior $p(\mathbf{z})$ and then penalize the divergence between $q_\phi(\mathbf{z})$ and $p(\mathbf{z})$ while optimizing for the above ELBO in Eq. (8). That is, we parameterize $p(\mathbf{z}) = \pi_1\,\mathbf{N}(\mathbf{z};\mathbf{c}_1,\gamma_1^2\boldsymbol{\Theta}^{-1}) + \ldots + \pi_b\,\mathbf{N}(\mathbf{z};\mathbf{c}_b,\gamma_b^2\boldsymbol{\Theta}^{-1})$ where $\pi_i \propto 2^{i/2}$ (see Condition **2**), which encodes the desired clustering structure. This is then reflected on the marginal posterior $q(\mathbf{z})$ via augmenting the above ELBO as,

$$\mathbf{L}_\alpha(\mathbf{x};\theta,\phi) \quad\triangleq\quad \mathbb{E}_{\mathbf{z}\sim q_\phi}\Big[\log p_\theta(\mathbf{x}|\mathbf{z})\Big] - \mathbb{KL}\Big(q_\phi(\mathbf{z}||\mathbf{x})||p(\mathbf{z})\Big) - \alpha\mathbb{KL}\Big(q(\mathbf{z})||p(\mathbf{z})\Big)\,, \qquad (9)$$

where the penalty term $\alpha\mathbb{KL}(q(\mathbf{z})||p(\mathbf{z}))$ serves as an incentive to encourage $q(\mathbf{z})$ to assume the same clustering structure as $p(\mathbf{z})$. The parameter $\alpha$ can be manually set to adjust the strength of the incentive. To encourage separation among learned clusters (see Condition **3**), we also add an extra penalty term to the above augmented ELBO,

$$\mathbf{L}_{\alpha,\beta}(\mathbf{x};\theta,\phi) \quad\triangleq\quad \mathbf{L}_\alpha(\mathbf{x};\theta,\phi) + \beta\sum_{i\neq j}\mathbb{KL}\Big(\mathbf{N}\left(\mathbf{z};\mathbf{c}_i,\gamma_i^2\boldsymbol{\Theta}^{-1}\mathbf{I}\right)\,||\,\mathbf{N}\left(\mathbf{z};\mathbf{c}_j,\gamma_j^2\boldsymbol{\Theta}^{-1}\mathbf{I}\right)\Big)\,. \qquad (10)$$

Once these clusters are learned, we can use the resulting encoding network $q_\phi(\mathbf{z}|\mathbf{x})$ to transform each training input $\mathbf{x}$ into its latent projection and subsequently train an SSGP on the latent space of $\mathbf{z}$ (instead of training it on the original data space). Our previous analysis can then be applied on $\mathbf{z}$ to give the desired sample complexity. The empirical efficiency of the proposed method is demonstrated in Section 4 below. Note that the cost of training the embedding is linear in the number of data points and therefore does not noticeably affect our overall running time.

## 4 Experiments

**Datasets.** This section presents our empirical studies on two real datasets: (a) the ABALONE dataset [42] with 3000 data points which was used to train a model that predicts the age of abalone (number of rings on its shell) from physical measurements such as length, diameter, height, whole weight, shucked weight, viscera weight and shell weight; and (b) the GAS SENSOR dataset with 4 million data points [5, 6] which was used to train a model that predicts the CO concentration (ppm) from measurements of humidity, temperature, flow rate, heater voltage and the resistant measures of 14 gas sensors.

In both settings, we compare our revised SSGP method with the traditional SSGP on both datasets to demonstrate its sample efficiency. In particular, our SSGP method is applied on the embedded space of data which was generated and configured using the auto-encoding method in Section 3.3.2 to approximately meet the aforementioned Conditions **1**-**3**.

The detailed parameterization of our entire algorithm[7] is provided in Appendix C. The traditional SSGP method on the other hand was applied directly to the data space. The prediction root-mean-square-error (RMSE) achieved by each method is reported at different sample complexities in Figure 1 below. All reported performances were averaged over 5 independent runs on a computing server with a Tesla K40 GPU with 12GB RAM.

**Results and Discussions.** It can be observed from the results that at all levels of sample complexity, the revised SSGP achieves substantially better performance than its vanilla SSGP counterpart. This

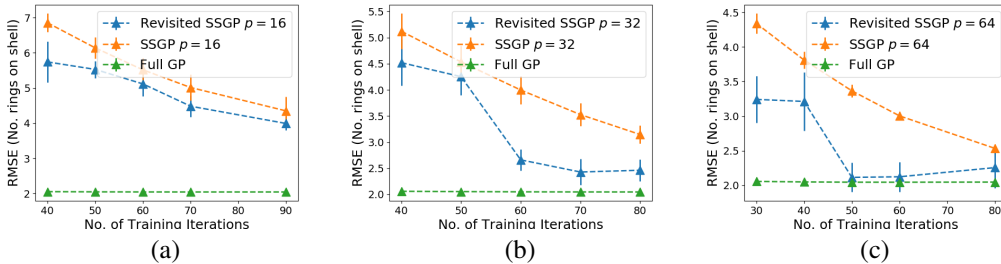

Figure 1: Graphs of performance comparisons between our revised SSGP and the traditional SSGP on the ABALONE dataset [42] at varying sample complexities (see Theorem 2) $p = 16, 32$ and $64$.

is expected since our revised SSGP is guaranteed to require many fewer samples than the vanilla SSGP when the data is reconfigured to exhibit a certain clustering structure (see Conditions **1**-**3** and Theorem 2). As such, when both are set to operate at the same level of sample complexity, one would expect the revised SSGP to achieve better performance since SSGP generally performs better when its sample complexity is set closer to the required threshold. On the larger GAS SENSOR dataset (which contains approximately $4M$ data points), we also observe the same phenomenon from the performance comparison graph as shown in Figure 2a below: A vanilla SSGP needs to increase its number of samples to marginally improve its predictive performance while our revisited SSGP is able to outperform the former with the least number of samples ($p = 16$).

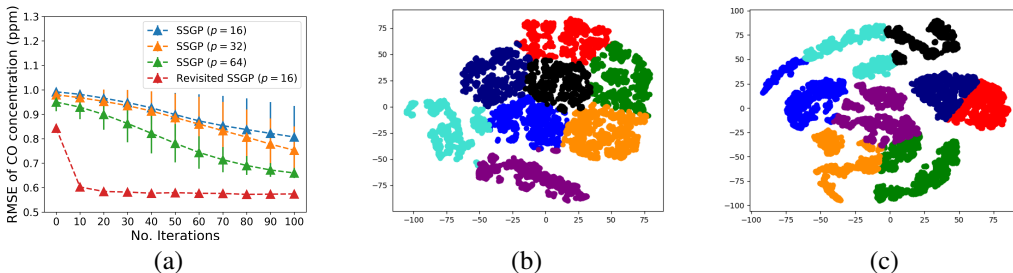

Figure 2: Graphs of (a) performance comparison between our revisited SSGP's (with sample complexity $p = 16$) and the vanilla SSGP's (with sample complexity $p = 16, 32, 64$) on the GAS SENSOR dataset [4]; and visualizations of (b) original and (c) reconfigured data distributions of GAS SENSOR data on a 2-dimensional latent space generated by our auto-encoding algorithm in Section 3.3.

Furthermore, a closer look into the data distribution (visualized on a 2D space in Fig. 2b) and the data reconfigured data distribution (visualized on a 2D space in Fig. 2c) also corroborates our hypothesis earlier that a well-separated data partition with high in-cluster concentration (in the form of a mixture of clusters – see Condition **1**) can be found (by our embedding algorithm in Section 3.3) to reconfigure our data distribution to (approximately) meet the necessary technical conditions that enable our sample-complexity enhancement analysis (see Section 3.1). Due to limited space, interested readers are referred to Appendix D for more detailed empirical studies and demonstrations.

## 5 Conclusion

We present a new method and analysis for approximating Gaussian processes. We obtain provable guarantees for both training and inference, which are the first to hold simultaneously over the entire space of kernel parameters. Our results complement existing work in kernel approximation that often assumes knowledge of its defining parameters. Our results also reveal important (practical) insights that allow us to develop an algorithmic handle on the tradeoff between approximation quality and sample complexity, which is achieved via finding an embedding that disentangles the latent coordinates of data. Our empirical results show for many datasets, such a disentangled embedding space can be found, which leads to a significantly reduced sample complexity of SSGP.

# 6  Statement of Broader Impact

Our work focuses on approximating Gaussian processes using a mixture of practical methods and theoretical analysis to reconfigure data in ways that reduce their approximation complexity. As such, it could have significant broader impact by allowing users to more accurately solve practical problems such as the ones discussed in our introduction, while still providing concrete theoretical guarantees. While applications of our work to real data could result in ethical considerations, this is an indirect (and unpredictable) side-effect of our work. Our experimental work uses publicly available datasets to evaluate the performance of our algorithms; no ethical considerations are raised.

# 7  Acknowledgement

T. N. Hoang is supported by the MIT-IBM Watson AI Lab, IBM Research. Q. M. Hoang is supported by the Gordon and Betty Moore Foundation's Data-Driven Discovery Initiative through Grant GBMF4554, by the US National Science Foundation (DBI-1937540), by the US National Institutes of Health (R01GM122935), and by the generosity of Eric and Wendy Schmidt by recommendation of the Schmidt Futures program. D. Woodruff is supported by National Institute of Health grant 5R01 HG 10798-2, Office of Naval Research grant N00014-18-1-2562, and a Simons Investigator Award.

## Footnotes

[1]For simplicity, we assume a zero mean function since we can always re-center the training outputs around 0.

[2]See Theorem 6.28 in Chapter 6 of [31].

[3]In contrast, existing literature often generates bounds on either an SSGP's prediction or its model evidence (for training) for a single parameter configuration, which makes such an analysis only heuristic.

[4]The intuition here is that kernel entries in narrower bands are cheaper (in term of sample cost) to approximate.

[5]The exact bounds defining the band can be found in Appendix A.

[6][3, 32] achieves a complexity of $\mathbf{O}(nm^2)$ where $m$ scales with the effective dimension of the kernel matrix.

[7]Our experimental code is released at `https://github.com/hqminh/gp_sketch_nips`.

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
