[Supplementary Material]

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

[8]This simplifies the analysis and does not restrict the expressiveness of the kernel since we can either normalize the output or absorb it into the length-scales (i.e., the $\theta_i$).

[9]The entire GAS SENSOR dataset contains approximately 4M data points. However, in the body of this paper we only used a sample of $500$K points to conduct our experiments.

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

# A  Intermediate Results for Theorem 2

Let $\Delta(\mathbf{x}_u, \mathbf{x}_v) \triangleq |\mathbf{K}(\mathbf{x}_u, \mathbf{x}_v) - \mathbf{K}'(\mathbf{x}_u, \mathbf{x}_v)|$ where $\mathbf{K}'(\mathbf{x}_u, \mathbf{x}_v) = (1/p) \sum_{i=1}^{p} \mathbf{K}_{\epsilon_i}(\mathbf{x}_u, \mathbf{x}_v)$, and where $\epsilon_i \sim \mathbf{N}(\mathbf{0}, \mathbf{I})$ as defined in Lemma 1 above. We will first measure the approximation loss across different value-bands of $\mathbf{K}(\mathbf{x}_u, \mathbf{x}_v)$, thereby deriving tight sample bounds for each band. Combining these with the union bound allows us to establish a much cheaper overall sample complexity as compared to the naïve $\mathbf{O}(n^2 \log n)$ bound.

**Lemma 2.** *Suppose the data distribution follows Conditions* **1**-**3** *above. Let* $\mathbf{c}(\mathbf{x}_u)$ *denote the cluster index of each data point* $\mathbf{x}_u$. *Let* $\mathcal{C} \triangleq \{u, v \mid \mathbf{c}(\mathbf{x}_u) = \mathbf{c}(\mathbf{x}_v)\}$ *and* $\mathcal{C}' \triangleq \{u, v \mid \mathbf{c}(\mathbf{x}_u) \neq \mathbf{c}(\mathbf{x}_v)\}$ *denote the sets of in-cluster and out-cluster kernel entries, respectively, where* $|\mathcal{C}| \simeq \frac{n^2}{4}$ *and* $|\mathcal{C}'| \simeq \frac{3n^2}{4}$.

*Proof.* By Condition **2**, since $n$ data points are scattered across $b$ clusters and each cluster $i$ has $2^{i/2}$ points, it follows that:

$$n = \sum_{i=1}^{b} 2^{\frac{i}{2}} = \frac{\sqrt{2^{b+1}} - \sqrt{2}}{\sqrt{2} - 1}$$

$$\Rightarrow |\mathcal{C}| = \sum_{i=1}^{b} 2^i = 2^{b+1} - 1 = \left(n\left(\sqrt{2} - 1\right) + \sqrt{2}\right)^2 - 1 \simeq \frac{n^2}{4}$$

$$\Rightarrow |\mathcal{C}'| = n^2 - |\mathcal{C}| \simeq \frac{3n^2}{4} . \tag{11}$$

This also implies that $b = \mathbf{O}(\log n)$, which is consistent with Condition **1** above. $\qquad\square$

**Lemma 3.** *Let* $\mathcal{C}_i = \{(u, v) \in \mathcal{C} \mid \mathbf{c}(\mathbf{x}_u) = \mathbf{c}(\mathbf{x}_v) = i\}$ *for* $i \in [1 \dots b]$. *Then with probability at least* $1 - \delta$, *for* $\delta \geq \mathbf{O}(\exp(\log n - \sqrt{d}))$, *the following holds for all* $i$ *and* $(u, v) \in \mathcal{C}_i$ *for which* $u \neq v$:

$$\left(1 - \frac{1}{2^{a+i-1}}\right)^{\frac{1}{4}} \leq \mathbf{K}(\mathbf{x}_u, \mathbf{x}_v) < \left(1 - \frac{1}{2^{a+i}}\right)^{\frac{1}{4}} \text{ where } a = \frac{1}{\log 2} \log\left(\frac{n^4}{n^4 - \lambda^4}\right) \tag{12}$$

*Proof.* If $\mathbf{x}_u$ and $\mathbf{x}_v$ are both generated from component $i$ of the data distribution as defined in Condition **1**, it follows that $\mathbf{\Theta}^{-1/2}(\mathbf{x}_u - \mathbf{x}_v) \sim \mathbf{N}(\mathbf{c}_i, \gamma_i^2 \mathbf{I})$. Therefore, by standard chi-squared tail bounds, with probability at least $1 - 2e^{-t}$, we have:

$$\left\|\mathbf{\Theta}^{-1/2}(\mathbf{x}_u - \mathbf{x}_v)\right\|_2^2 = \gamma_i^2 d \pm \mathbf{O}\left(\gamma_i^2 \sqrt{dt}\right), \tag{13}$$

where $d$ is the data dimension. Using this, we can then figure out a setting for $\gamma_i^2$ such that $\mathbf{K}(\mathbf{x}_u, \mathbf{x}_v)$ follows the above condition in Eq. 12. In particular, set

$$\mathcal{L}(i) = \log\left(\frac{2^{a+i}}{2^{a+i} - 1}\right) \quad \text{and} \quad \mathcal{U}(i) = \log\left(\frac{2^{a+i-1}}{2^{a+i-1} - 1}\right). \tag{14}$$

We can then choose:

$$\gamma_i^2 = \frac{1}{4d}\left(\mathcal{U}(i) + \mathcal{L}(i)\right) \quad \text{and} \quad t = \sqrt{d}\left(\frac{\mathcal{U}(i) - \mathcal{L}(i)}{\mathcal{U}(i) + \mathcal{L}(i)}\right) \simeq \mathbf{O}\left(\sqrt{d}\right), \tag{15}$$

so that by plugging these choices in Eq. 13 above, we have with probability at least $1 - 2e^{-t}$:

$$\left\|\mathbf{\Theta}^{-1/2}(\mathbf{x}_u - \mathbf{x}_v)\right\|_2^2 \in \left[\frac{1}{2}\log\left(\frac{2^{a+i}}{2^{a+i} - 1}\right), \frac{1}{2}\log\left(\frac{2^{a+i-1}}{2^{a+i-1} - 1}\right)\right]$$

$$\Rightarrow \mathbf{K}(\mathbf{x}_u, \mathbf{x}_v) \in \left[\left(1 - \frac{1}{2^{a+i-1}}\right)^{\frac{1}{4}}, \left(1 - \frac{1}{2^{a+i}}\right)^{\frac{1}{4}}\right]. \tag{16}$$

Now, note that for any $\delta$ for which $\delta \geq \mathbf{O}\left(4^b e^{-\sqrt{d}}\right) \geq \mathbf{O}\left(4^i e^{-\sqrt{d}}\right) \forall i \leq b$, we have $\delta/4^i \geq 2e^{-t}$ since $t \simeq \mathbf{O}(\sqrt{d})$. This also means $\delta \geq \mathbf{O}(\exp(\log n - \sqrt{d}))$ since $b = \mathbf{O}(\log n)$.

That is, Eq. (16) and hence, Eq. (12), hold with probability at least $1 - 2e^{-t} \geq 1 - \delta/4^i$ for each entry in $\mathcal{C}_i$. For each cluster $i$, even though there are up to $2^i$ kernel entries, by the triangle inequality it is easy to see that we only need to apply a union bound over at most $2^{i/2}$ (carefully selected) entries (excluding the entries on the diagonal) to meet Eq. (12) with probability at least $1 - 2^i(\delta/4^i) = 1 - \delta/2^i$.

Subsequently, applying a union bound over all clusters gives us that with probability at least $1 - \delta \sum_{i=1}^{b} 1/2^i \geq 1 - \delta$, all kernel entries within the $i$-th cluster satisfy Eq. (12) simultaneously for $1 \leq i \leq b$. $\qquad\square$

**Lemma 4.** *For all* $(u, v) \in \mathcal{C}' \triangleq \{u, v \mid \mathbf{c}(\mathbf{x}_u) \neq \mathbf{c}(\mathbf{x}_v)\}$, *we have* $\mathbf{K}(\mathbf{x}_u, \mathbf{x}_v) < \left(1 - \dfrac{1}{2^a}\right)^{\frac{1}{4}}$ *where* $a = \dfrac{1}{\log 2} \log\left(\dfrac{n^4}{n^4 - \lambda^4}\right)$ *as defined in Lemma 3 above.*

*Proof.* For any $(u, v)$ for which $\mathbf{c}(\mathbf{x}_u) = i$ and $\mathbf{c}(\mathbf{x}_v) = j$ and $i \neq j$, we have:

$$
\begin{aligned}
\left\|\mathbf{\Theta}^{-1/2}(\mathbf{x}_u - \mathbf{x}_v)\right\|_2^2 &\geq \left\|\mathbf{\Theta}^{-1/2}(\mathbf{c}_i - \mathbf{c}_j)\right\|_2^2 - \left\|\mathbf{\Theta}^{-1/2}(\mathbf{x}_u - \mathbf{c}_i)\right\|_2^2 - \left\|\mathbf{\Theta}^{-1/2}(\mathbf{x}_v - \mathbf{c}_j)\right\|_2^2 \\
&\geq \left\|\mathbf{\Theta}^{-1/2}(\mathbf{c}_i - \mathbf{c}_j)\right\|_2^2 - \frac{1}{2}\log\left(\frac{2^{a+i}}{2^{a+i} - 1}\right) - \frac{1}{2}\log\left(\frac{2^{a+j}}{2^{a+j} - 1}\right) \\
&\geq \left\|\mathbf{\Theta}^{-1/2}(\mathbf{c}_i - \mathbf{c}_j)\right\|_2^2 - \log\left(\frac{2^a}{2^a - 1}\right) \\
\Rightarrow \mathbf{K}(\mathbf{x}_u, \mathbf{x}_v) &= \exp\left(-\frac{1}{2}\left\|\mathbf{\Theta}^{-1/2}(\mathbf{x}_u - \mathbf{x}_v)\right\|_2^2\right) \\
&\leq \exp\left(-\frac{1}{2} \cdot \left\|\mathbf{\Theta}^{-1/2}(\mathbf{c}_i - \mathbf{c}_j)\right\|_2^2 + \frac{1}{2}\log\left(\frac{2^a}{2^a - 1}\right)\right) < \left(1 - \frac{1}{2^a}\right)^{\frac{1}{4}}
\end{aligned}
$$

since for all $(i, j)$, by Condition **3**:

$$
\left\|\mathbf{\Theta}^{-1/2}(\mathbf{c}_i - \mathbf{c}_j)\right\|_2^2 > \frac{3}{2}\log\left(\frac{2^a}{2^a - 1}\right) . \tag{17}
$$

This completes our proof for the stated result of Lemma 4. $\qquad\square$

**Corollary 1.** *With probability at least* $1 - \delta$, *there are exactly* $n$ *entries that are greater than* $1 - 2^{-(a+b)}$ *where* $a = \frac{1}{\log 2}\log\left(\frac{n^4}{n^4 - \lambda^4}\right)$. *These are the diagonal entries* $\mathbf{K}(\mathbf{x}_u, \mathbf{x}_u)$ *with* $1 \leq u \leq n$.

*Proof.* Lemma 3 asserts that with probability $1 - \delta$, all kernel entries $\mathbf{K}(\mathbf{x}_u, \mathbf{x}_v)$, where $\mathbf{c}(\mathbf{x}_u) = \mathbf{c}(\mathbf{x}_v) = i$, belong to their respective band $\kappa_i = \{(u, v) \mid 1 - 1/2^{a+i-1} \leq \mathbf{K}^4(\mathbf{x}_u, \mathbf{x}_v) \leq 1 - 1/2^{a+i}\}$. When this happens, all in-cluster entries (except the diagonal entries) will have values between $1 - 1/2^a$ and $1 - 1/2^{a+b}$ (since there are $b$ bands) and as such, off-cluster entries will either be smaller than $1 - 1/2^a$ or larger than $1 - 1/2^{a+b}$. But then Lemma 4 further guarantees that all off-cluster entries are smaller than $1 - 1/2^a$, following Condition **3**. Thus, it follows that the only entries that are larger than $1 - 1/2^{a+b}$ are the diagonal items and there are exactly $n$ of them. $\qquad\square$

**Lemma 5.** *Let* $\kappa_i = \{(u, v) \mid 1 - 1/2^{a+i-1} \leq \mathbf{K}^4(\mathbf{x}_u, \mathbf{x}_v) < 1 - 1/2^{a+i}\}$. *It follows that for each* $i \in [1 \ldots b]$, *with probability at least* $1 - \delta/b$:

$$
\sum_{(u,v) \in \mathcal{G}_i} \Delta^2(\mathbf{x}_u, \mathbf{x}_v) \leq \frac{\lambda^2}{b} , \tag{18}
$$

*if the kernel approximation* $\mathbf{K}'(\mathbf{x}_u, \mathbf{x}_v) \triangleq \frac{1}{p}\sum_{t=1}^{p} \mathbf{K}_{\epsilon_t}(\mathbf{x}_u, \mathbf{x}_v)$ *is formed using at least* $p = \dfrac{b|\kappa_i|}{\lambda^2 \cdot 2^{a+i}}\log\left(\dfrac{b|\kappa_i|}{\delta}\right) = \mathbf{O}\left(\dfrac{\log^2 n}{\lambda^2}\log\left(\dfrac{\log n}{\delta}\right)\right)$ *samples.*

*Proof.* For all $(u, v)$, we have $\mathbf{K}_{\boldsymbol{\epsilon}_t}(\mathbf{x}_u, \mathbf{x}_v) = \cos(\boldsymbol{\epsilon}_t^\top \boldsymbol{\Theta}^{-1/2}(\mathbf{x}_u - \mathbf{x}_v))$ where $\boldsymbol{\epsilon}_t \sim \mathbf{N}(0, \mathbf{I})$ and,

$$\mathbf{K}_{\boldsymbol{\epsilon}_t}(\mathbf{x}_u, \mathbf{x}_v) = \cos\left(\sum_{\ell=1}^{d} \epsilon_t^\ell \cdot \left(\frac{\mathbf{x}_u^\ell - \mathbf{x}_v^\ell}{\theta_\ell}\right)\right) \triangleq \cos\left(\mathbf{z}_{uv}^t\right) . \tag{19}$$

Since $\epsilon_t^\ell \sim \mathbf{N}(0, 1)$, $\mathbf{z}_{uv}^t$ is then a weighted sum of Gaussian random variables and $\mathbf{z}_{uv}^t \sim \mathbf{N}(0, \boldsymbol{\Sigma}_{uv}^t)$, where $\boldsymbol{\Sigma}_{uv}^t \triangleq (\mathbf{x}_u - \mathbf{x}_v)^\top \boldsymbol{\Theta}^{-1}(\mathbf{x}_u - \mathbf{x}_v)$, which in turn implies:

$$\mathbb{E}[\cos(\mathbf{z}_{uv}^t)] = \exp\left(-0.5\boldsymbol{\Sigma}_{uv}^t\right) = \mathbf{K}(\mathbf{x}_u, \mathbf{x}_v) ,$$

$$\mathbb{V}[\cos(\mathbf{z}_{uv}^t)] = \frac{1}{2}\left[1 - \mathbb{E}[\cos(\mathbf{z}_{uv}^t)]^2\right]^2 = \frac{1}{2}\left(1 - \mathbf{K}^2(\mathbf{x}_u, \mathbf{x}_v)\right)^2 \leq 2 \times \frac{1}{2^{a+i}} , \tag{20}$$

where the last inequality follows from the choice of $(u, v) \in \kappa_i$ and the definition of the $\kappa_i$ above. Next, applying the Chernoff-Hoeffding inequality and union bounding over the $\kappa_i$, we have:

$$\Pr\left(\forall (u, v) \in \kappa_i : \Delta(\mathbf{x}_u, \mathbf{x}_v) \leq \frac{\epsilon}{p}\right) \geq 1 - 2|\kappa_i|\exp\left(-\frac{\epsilon^2}{4\sum_{t=1}^{p} \mathbb{V}[\cos(\mathbf{z}_{uv}^t)]}\right)$$

$$\Rightarrow \Pr\left(\sum_{(u,v)\in\kappa_i} \Delta^2(\mathbf{x}_u, \mathbf{x}_v) \leq \frac{|\kappa_i|\epsilon^2}{p^2}\right) \geq 1 - 2|\kappa_i|\exp\left(-\frac{\epsilon^2 \cdot 2^{a+i}}{8p}\right) . \tag{21}$$

Thus, setting $\epsilon^2 = \frac{\lambda^2 p^2}{4b|\kappa_i|}$ and $p \geq \frac{32b|\kappa_i|}{\lambda^2 \cdot 2^{a+i}} \log\left(\frac{2b|\kappa_i|}{\delta}\right)$ yields:

$$\Pr\left(\sum_{(u,v)\in\mathcal{G}_i} \Delta^2(\mathbf{x}_u, \mathbf{x}_v) \leq \frac{\lambda^2}{4b}\right) \geq 1 - 2|\kappa_i|\exp\left(-\frac{\lambda^2 p \cdot 2^{a+i}}{32b|\kappa_i|}\right) \geq 1 - \frac{\delta}{b} . \tag{22}$$

where the last inequality follows from the above choice of $p$. Since $|\kappa_i| = 2^i$ by Condition **2**, we further have $p \geq \frac{32b}{\lambda^2 \cdot 2^a} \log\left(\frac{b \cdot 2^{b+1}}{\delta}\right) = \mathbf{O}\left(\frac{\log^2 n}{\lambda^2} \log\left(\frac{\log n}{\delta}\right)\right)$. $\qquad \square$

Lemma 5 thus establishes a very strong sample complexity of $\mathbf{O}(\log^2 n \log \log n)$ for approximating all kernel entries within a narrow band of values, which is significantly cheaper than the sample complexity of $\mathbf{O}(n^2 \log n)$ we would get if we were to ignore the distribution of kernel values in different bands. This is made clear in Corollary 2 below, which combines Lemmas 3, 4 and 5 to establish an overall sample complexity resulting in only a small approximation loss accumulated over all bands.

**Corollary 2.** *If a kernel approximation $\mathbf{K}'$ of $\mathbf{K}$ is formed such that $\mathbf{K}'(\mathbf{x}_u, \mathbf{x}_v) \triangleq \frac{1}{p}\sum_{t=1}^{p} \mathbf{K}_{\boldsymbol{\epsilon}_t}(\mathbf{x}_u, \mathbf{x}_v)$ for all in-cluster entries $(u, v) \in \mathcal{C}$ using $p = \mathbf{O}\left((\log^2 n/\lambda^2) \log (\log n/\delta)\right)$ samples and $\mathbf{K}'(\mathbf{x}_{u'}, \mathbf{x}_{v'}) \triangleq 0$ for all off-cluster entries $(u', v') \in \mathcal{C}'$, then,*

$$\|\mathbf{K} - \mathbf{K}'\|_2^2 \leq \|\mathbf{K} - \mathbf{K}'\|_F^2 \leq \lambda^2 ,$$

*with probability at least $1 - 2\delta$ with $\delta \geq \mathbf{O}(\exp(\log n - \sqrt{d}))$. This immediately guarantees that $\mathbf{K}'$ is spectrally close to $\mathbf{K}$ using the notion of $\lambda$-closeness (see Definition 1).*

*Proof.* By Lemma 3, with probability $1 - \delta$, $|\kappa_i| = |\mathcal{C}_i|$ simultaneously for all $i$. Thus, applying a union bound over this event and the results obtained in Lemma 5 for all clusters, we have the following bound on the total approximation loss over in-cluster entries in $\mathcal{C}$ with probability $1 - 2\delta$:

$$\sum_{(u,v)\in\mathcal{C}} \Delta^2(\mathbf{x}_u, \mathbf{x}_v) \leq \frac{\lambda^2}{4} . \tag{23}$$

Furthermore, by Lemma 4, we also have the following bound on the total approximation loss over off-cluster entries in $\mathcal{C}'$ (which were approximated uniformly by zero):

$$\sum_{(u,v)\in\mathcal{C}'} \Delta^2(\mathbf{x}_u, \mathbf{x}_v) \leq \frac{3n^2}{4}\left(\mathbf{K}(\mathbf{x}_u, \mathbf{x}_v) - 0\right)^2 \leq \frac{3n^2}{4}\sqrt{1 - \frac{1}{2^a}} = \frac{3\lambda^2}{4} , \tag{24}$$

when the last inequality is due to the facts (established in Lemma 4) that $\mathbf{K}^4(\mathbf{x}_u, \mathbf{x}_v) \leq 1 - 1/2^a$ and that $a = \dfrac{1}{\log 2} \log \left( \dfrac{n^4}{n^4 - \lambda^4} \right)$. Finally, combining these yields:

$$\|\mathbf{K} - \mathbf{K}'\|_2^2 \leq \|\mathbf{K} - \mathbf{K}'\|_F^2 = \sum_{(u,v) \in \mathcal{C}} \Delta^2(\mathbf{x}_u, \mathbf{x}_v) + \sum_{(u,v) \in \mathcal{C}'} \Delta^2(\mathbf{x}_u, \mathbf{x}_v) \leq \frac{1}{4}\lambda^2 + \frac{3}{4}\lambda^2 = \lambda^2 \tag{25}$$

$\square$

## B  Intermediate Results for Theorem 3

**Lemma 6.** *Let $\mathbf{K}$ and $\mathbf{K}'$ be positive semidefinite matrices in $\mathbb{R}^{n \times n}$ such that $-\lambda\mathbf{I} \preceq \mathbf{K} - \mathbf{K}' \preceq \lambda\mathbf{I}$, $\mathbf{Q} \triangleq \mathbf{K} + \sigma^2\mathbf{I}$ and $\mathbf{Q}' \triangleq \mathbf{K}' + \sigma^2\mathbf{I}$ for some $\lambda, \sigma > 0$, then:*

$$\|\mathbf{Q}'^{-1}\|_2 = \left( 1 \pm \frac{\lambda}{\sigma^2} \right) \|\mathbf{Q}^{-1}\|_2 . \tag{26}$$

*Proof.* By definition of the spectral norm, we have $\forall \mathbf{x} \in \mathbb{R}^n$:

$$\mathbf{K} - \mathbf{K}' \preceq \lambda\mathbf{I}, \tag{27}$$

which implies

$$\begin{aligned} \mathbf{Q} &\preceq \mathbf{K}' + (\sigma^2 + \lambda)\mathbf{I} \\ &\preceq (\sigma^2 + \lambda)\mathbf{K}' + (\sigma^2 + \lambda)\mathbf{I} \\ &= \left( 1 + \frac{\lambda}{\sigma^2} \right) \mathbf{Q}' . \end{aligned} \tag{28}$$

where $\preceq$ and $\succeq$ denote the Loewner inequality operators. Likewise, by symmetry, we also have:

$$\mathbf{Q}' \preceq \left( 1 + \frac{\lambda}{\sigma^2} \right) \mathbf{Q} . \tag{29}$$

Let $\mathbf{A} \triangleq (1 + \lambda/\sigma^2)\mathbf{Q}'$ and $\mathbf{B} \triangleq \mathbf{Q}$. Since $\mathbf{A}$ and $\mathbf{B}$ are symmetric and positive semidefinite, there exist $\mathbf{U}, \mathbf{V}$ with orthogonal rows and columns and diagonal matrices $\mathbf{\Sigma}, \mathbf{\Sigma}'$ for which $\mathbf{A} = \mathbf{U}\mathbf{\Sigma}\mathbf{U}^\top$ and $\mathbf{B} = \mathbf{V}\mathbf{\Sigma}'\mathbf{V}^\top$. We further let $\mathbf{A}^{-1/2} \triangleq \mathbf{U}\mathbf{\Sigma}^{-1/2}$ and $\mathbf{B}^{-1/2} \triangleq \mathbf{V}\mathbf{\Sigma}'^{-1/2}$.

Then, we can rewrite Eq. (28) as:

$$\begin{aligned} \mathbf{A} - \mathbf{B} &\succeq 0 \\ \Rightarrow \mathbf{B}^{-1/2}(\mathbf{A} - \mathbf{B})\mathbf{B}^{-1/2} &\succeq 0 \\ \Rightarrow \mathbf{B}^{-1/2}\mathbf{A}\mathbf{B}^{-1/2} - \mathbf{I} &\succeq 0 \\ \Rightarrow \mathbf{A}^{-1/2}\mathbf{B}^{1/2}(\mathbf{B}^{-1/2}\mathbf{A}\mathbf{B}^{-1/2})\mathbf{B}^{-1/2}\mathbf{A}^{1/2} &\succeq \mathbf{A}^{-1/2}\mathbf{B}^{1/2}\mathbf{B}^{-1/2}\mathbf{A}^{1/2} \\ \Rightarrow \mathbf{A}^{1/2}\mathbf{B}^{-1}\mathbf{A}^{1/2} &\succeq \mathbf{I} \\ \Rightarrow \mathbf{A}^{-1/2}(\mathbf{A}^{1/2}\mathbf{B}^{-1}\mathbf{A}^{1/2})\mathbf{A}^{-1/2} &\succeq \mathbf{A}^{-1/2}\mathbf{A}^{-1/2} \\ \Rightarrow \mathbf{B}^{-1} &\succeq \mathbf{A}^{-1} \\ \Rightarrow \mathbf{Q}^{-1} &\succeq \frac{\sigma^2}{\sigma^2 + \lambda}\mathbf{Q}'^{-1} \\ \Rightarrow \left( 1 + \frac{\lambda}{\sigma^2} \right) \mathbf{Q}^{-1} &\succeq \mathbf{Q}'^{-1} . \end{aligned} \tag{30}$$

Again, by symmetry, we can rewrite Eq. 29 as:

$$\begin{aligned} \mathbf{Q}'^{-1} &\succeq \frac{\sigma^2}{\sigma^2 + \lambda}\mathbf{Q}^{-1} \succeq \left( 1 - \frac{\lambda}{\sigma^2 + \lambda} \right) \mathbf{Q}^{-1} \\ &\succeq \left( 1 - \frac{\lambda}{\sigma^2} \right) \mathbf{Q}^{-1} . \end{aligned} \tag{31}$$

Therefore, we have $\|\mathbf{Q}'^{-1}\|_2 = (1 \pm \lambda/\sigma^2)\|\mathbf{Q}^{-1}\|_2$ . $\square$

Let $g(\mathbf{x}_*)$ and $g'(\mathbf{x}_*)$ respectively denote the predictive distributions of full GP and the approximated GP pertaining to an arbitrary test input $\mathbf{x}_*$. We then state the following lemmas:

**Lemma 7.** *Let $\mathbf{K}'$ denote an approximation that is $\lambda$-close to the original kernel $\mathbf{K}$. The induced predictive mean of $\mathbf{K}'$ is bounded by a factor of $1 \pm \lambda/\sigma^2$ times the original predictive mean.*

$$\mathbb{E}[g(\mathbf{x}_*)] = \left(1 \pm \frac{\lambda}{\sigma^2}\right)\mathbb{E}[g'(\mathbf{x}_*)] . \tag{32}$$

*Proof.* Let $\mathbf{k}_* \triangleq [k(\mathbf{x}_*, \mathbf{x}_i)]_{i=1}^n$ where $\mathbf{x}_i$ denotes the $i$-th training data point. We have:

$$
\begin{aligned}
\mathbb{E}[g(\mathbf{x}_*)] &= \frac{1}{2}\left((\mathbf{k}_* + \mathbf{y})^\top \mathbf{Q}^{-1}(\mathbf{k}_* + \mathbf{y}) - \mathbf{k}_*^\top \mathbf{Q}^{-1}\mathbf{k}_* - \mathbf{y}^\top \mathbf{Q}^{-1}\mathbf{y}\right) \\
&= \frac{1}{2}\left(1 \pm \frac{\lambda}{\sigma^2}\right)\left((\mathbf{k}_* + \mathbf{y})^\top \mathbf{Q}'^{-1}(\mathbf{k}_* + \mathbf{y}) - \mathbf{k}_*^\top \mathbf{Q}'^{-1}\mathbf{k}_* - \mathbf{y}^\top \mathbf{Q}'^{-1}\mathbf{y}\right) \\
&= \left(1 \pm \frac{\lambda}{\sigma^2}\right)\mathbb{E}[g'(\mathbf{x}_*)] ,
\end{aligned}
\tag{33}
$$

where the first and third equations follow from adding and subtracting the same terms to the expression of $g(\mathbf{x}_*)$ – see Eq. (2) – while the second equation follows from applying Lemma 6 above. $\square$

**Lemma 8.** *Let $\mathbf{K}'$ denote an approximation that is $\lambda$-close to the original kernel $\mathbf{K}$. The induced predictive variance of $\mathbf{K}'$ is bounded by a factor of $1 \pm \lambda/\sigma^2$ of the original predictive variance up to a constant bias of $\lambda/\sigma^2$,*

$$\mathbb{V}[g(\mathbf{x}_*)] = \left(1 \pm \frac{\lambda}{\sigma^2}\right)\mathbb{V}[g'(\mathbf{x}_*)] \pm \frac{\lambda}{\sigma^2} . \tag{34}$$

*Proof.* Following the definition of the Gaussian kernel, we assume that the signal of the SE (Squared Exponential) kernel is unitary [8]. As such,

$$
\begin{aligned}
\mathbb{V}[g(\mathbf{x}_*)] &= 1 - \mathbf{k}_*^\top \mathbf{Q}^{-1}\mathbf{k}_* \\
&= 1 - \left(1 \pm \frac{\lambda}{\sigma^2}\right)\mathbf{k}_*^\top \mathbf{Q}'^{-1}\mathbf{k}_* \\
&= \left(1 \pm \frac{\lambda}{\sigma^2}\right)\left(1 - \mathbf{k}_*^\top \mathbf{Q}'^{-1}\mathbf{k}_*\right) \pm \frac{\lambda}{\sigma^2} \\
&= \left(1 \pm \frac{\lambda}{\sigma^2}\right)\mathbb{V}[g'(\mathbf{x}_*)] \pm \frac{\lambda}{\sigma^2} ,
\end{aligned}
\tag{35}
$$

where (again) the above equation follows straightforwardly from applying Lemma 6 and standard algebraic manipulation. Lemma 7 and Lemma 8 thus provide an explicit bound on the difference between the original and approximated predictive distributions. We will now establish another bound on the difference between the original and approximated negative log likelihoods (i.e., the training objectives) in Lemma 9 and Lemma 10 below. $\square$

**Lemma 9.** *Let $\mathbf{K}'$ denote an approximation that is $\lambda$-close to the original kernel $\mathbf{K}$. Let $\mathbf{Q} = \mathbf{K} + \sigma^2\mathbf{I}$ and $\mathbf{Q}' = \mathbf{K}' + \sigma^2\mathbf{I}$. We have:*

$$\log|\mathbf{Q}'| = \left(1 \pm \tau_{\lambda,\sigma}(\mathbf{K})\right)\log|\mathbf{Q}| . \tag{36}$$

*where the spectral constant $\tau_{\lambda,\sigma}(\mathbf{K})$ of $\mathbf{K}$ is defined below:*

$$
\tau_{\lambda,\sigma}(\mathbf{K}) \triangleq \frac{\max\left(\left|\log\left(1 + \frac{\lambda}{\sigma^2}\right)\right|, \left|\log\left(1 - \frac{\lambda}{\sigma^2}\right)\right|\right)}{\min\left(\left|\log(\lambda_{\min}(\mathbf{K}) + \sigma^2)\right|, \left|\log(\lambda_{\max}(\mathbf{K}) + \sigma^2)\right|\right)} . \tag{37}
$$

*Proof.* Let $\lambda_1 \leq \lambda_2 \cdots \leq \lambda_n$ and $\lambda_1' \leq \lambda_2' \cdots \leq \lambda_n'$ be the eigenvalues of $\mathbf{K}$ and $\mathbf{K}'$ respectively. Applying the Courant-Fischer theorem on the result obtained in Lemma 6, we have:

$$\lambda_i' + \sigma^2 = \left(1 \pm \frac{\lambda}{\sigma^2}\right)(\lambda_i + \sigma^2) . \tag{38}$$

This implies:

$$
\begin{aligned}
\log|\mathbf{Q}'| &\leq \sum_{i=1}^{n}\left|\log(\lambda_i' + \sigma^2)\right| = \sum_{i=1}^{n}\left|\log(\lambda_i + \sigma^2) + \log\left(1 \pm \frac{\lambda}{\sigma^2}\right)\right| \\
&\leq \sum_{i=1}^{n}\left|\log(\lambda_i + \sigma^2)\right| + \sum_{i=1}^{n}\max\left(\left|\log\left(1 + \frac{\lambda}{\sigma^2}\right)\right|, \left|\log\left(1 - \frac{\lambda}{\sigma^2}\right)\right|\right) \\
&\leq \left(1 + \tau_{\lambda,\sigma}(\mathbf{K})\right)\sum_{i=1}^{n}\left|\log(\lambda_i + \sigma^2)\right| = \left(1 + \tau_{\lambda,\sigma}(\mathbf{K})\right)\log|\mathbf{Q}| . 
\end{aligned}
\tag{39}
$$

Similarly, by symmetry, we have:

$$
\begin{aligned}
\log|\mathbf{Q}'| &\geq \sum_{i=1}^{n}\left|\log(\lambda_i + \sigma^2)\right| - \sum_{i=1}^{n}\max\left(\left|\log\left(1 + \frac{\lambda}{\sigma^2}\right)\right|, \left|\log\left(1 - \frac{\lambda}{\sigma^2}\right)\right|\right) \\
&\geq \left(1 - \tau_{\lambda,\sigma}(\mathbf{K})\right)\sum_{i=1}^{n}\left|\log(\lambda_i + \sigma^2)\right| = \left(1 - \tau_{\lambda,\sigma}(\mathbf{K})\right)\log|\mathbf{Q}| . 
\end{aligned}
\tag{40}
$$

Together, Eq. (39) and Eq. (40) imply $\log|\mathbf{Q}'| = \left(1 \pm \tau_{\lambda,\sigma}(\mathbf{K})\right)\log|\mathbf{Q}|$. $\qquad\square$

**Lemma 10.** *Let $\mathbf{K}'$ denote an approximation that is $\lambda$-close to the original kernel $\mathbf{K}$. With $\tau_{\lambda,\sigma}(\mathbf{K})$ previously defined in Lemma 9, we have:*

$$\ell'(\boldsymbol{\Theta}) = \left(1 \pm \max\left(\tau_{\lambda,\sigma}(\mathbf{K}), \frac{\lambda}{\sigma^2}\right)\right)\ell(\boldsymbol{\Theta}) . \tag{41}$$

*where $\ell(\boldsymbol{\Theta})$ and $\ell'(\boldsymbol{\Theta})$ respectively denote the negative log likelihood of the full GP and the approximated GP evaluated at the hyper-parameters $\boldsymbol{\Theta} = \mathrm{diag}[\theta_1^2, \theta_2^2 \ldots \theta_d^2]$ as defined previously.*

*Proof.* We have:

$$
\begin{aligned}
\ell'(\boldsymbol{\Theta}) &= \frac{1}{2}\log|\mathbf{Q}'| + \frac{1}{2}\mathbf{y}^\top\mathbf{Q}'^{-1}\mathbf{y} \\
&= \frac{1}{2}\left(1 \pm \tau_{\lambda,\sigma}(\mathbf{K})\right)\log|\mathbf{Q}| + \frac{1}{2}\left(1 \pm \frac{\lambda}{\sigma^2}\right)\mathbf{y}^\top\mathbf{Q}^{-1}\mathbf{y} \\
&= \left(1 \pm \max\left(\tau_{\lambda,\sigma}(\mathbf{K}), \frac{\lambda}{\sigma^2}\right)\right)\frac{1}{2}\left(\log|\mathbf{Q}| + \mathbf{y}^\top\mathbf{Q}^{-1}\mathbf{y}\right) \\
&= \left(1 \pm \max\left(\tau_{\lambda,\sigma}(\mathbf{K}), \frac{\lambda}{\sigma^2}\right)\right)\ell(\boldsymbol{\Theta}) . \qquad\square
\end{aligned}
\tag{42}
$$

Using the result of Lemma 10 above, we can further analyze how the quality of the optimized parameter $\boldsymbol{\Theta}_*' = \arg\max_{\boldsymbol{\Theta}} \ell'(\boldsymbol{\Theta})$ of the approximated training objective compares to the true optimizer of the original objective function $\boldsymbol{\Theta}_* = \arg\max_{\boldsymbol{\Theta}} \ell(\boldsymbol{\Theta})$ in Lemma 11 below.

**Lemma 11.** *Let $\boldsymbol{\Theta}_*$ and $\boldsymbol{\Theta}_*'$ denote the optimal hyper-parameters obtained by respectively minimizing the negative log likelihood of the full GP and the approximated GP. We have:*

$$\ell'(\boldsymbol{\Theta}_*') = \left(1 \pm \max\left(\tau_{\lambda,\sigma}(\mathbf{K}), \frac{\lambda}{\sigma^2}\right)\right)\ell(\boldsymbol{\Theta}_*) . \tag{43}$$

*Proof.* By Lemma 10, we have:

$$
\begin{aligned}
\ell'(\boldsymbol{\Theta}_*') &\leq \ell'(\boldsymbol{\Theta}_*) \\
&\leq \left(1 + \max\left(\tau_{\lambda,\sigma}(\mathbf{K}), \frac{\lambda}{\sigma^2}\right)\right)\ell(\boldsymbol{\Theta}_*)
\end{aligned}
\tag{44}
$$

and

$$\ell'(\mathbf{\Theta}'_*) \geq \left(1 - \max\left(\tau_{\lambda,\sigma}(\mathbf{K}), \frac{\lambda}{\sigma^2}\right)\right)\ell(\mathbf{\Theta}'_*)$$

$$\geq \left(1 - \max\left(\tau_{\lambda,\sigma}(\mathbf{K}), \frac{\lambda}{\sigma^2}\right)\right)\ell(\mathbf{\Theta}_*). \tag{45}$$

Together, these results imply $\ell'(\mathbf{\Theta}'_*) = \left(1 \pm \max\left(\tau_{\lambda,\sigma}(\mathbf{K}), \frac{\lambda}{\sigma^2}\right)\right)\ell(\mathbf{\Theta}_*)$. $\qquad\square$

**Lemma 12.** *Let $\delta \in (0,1)$ and let $\mathbf{K}'$ denote an approximation of $\mathbf{K}$ for which $\|\mathbf{K} - \mathbf{K}'\|_2^2 \leq \lambda^2$ with probability at least $1 - \delta$ uniformly over the entire parameter space. Let $\mathbf{\Theta}_*$ and $\mathbf{\Theta}'_*$ denote the optimal hyper-parameters obtained by respectively minimizing the negative log likelihood of the full GP and the approximated GP. Then, with probability $1 - \delta$, the following holds:*

$$\mathbb{E}[g'(\mathbf{x}_*; \mathbf{\Theta}'_*)] = \left(1 \pm \rho(\lambda, \sigma, \mathbf{\Theta}_*, \mathbf{\Theta}'_*)\right) \cdot \mathbb{E}[g(\mathbf{x}_*; \mathbf{\Theta}_*)] + \wp(\lambda, \sigma, \mathbf{\Theta}_*, \mathbf{\Theta}'_*) \tag{46}$$

*where $\rho(\lambda, \sigma, \mathbf{\Theta}_*, \mathbf{\Theta}'_*)$ and $\wp(\lambda, \sigma, \mathbf{\Theta}_*, \mathbf{\Theta}'_*)$ are constant with respect to $\lambda, \sigma, \mathbf{\Theta}_*, \mathbf{\Theta}'_*$*

*Proof.* We have:

$$\mathbb{E}[g(\mathbf{x}_*; \mathbf{\Theta}_*)] = \mathbf{k}_*^\top \mathbf{Q}^{-1} \mathbf{y}\Big|_{\mathbf{\Theta}_*}$$

$$= \frac{1}{2}\left[(\mathbf{k}_* + \mathbf{y})^\top \mathbf{Q}^{-1}(\mathbf{k}_* + \mathbf{y}) - \mathbf{k}_* \mathbf{Q}^{-1}\mathbf{k}_* + \log|\mathbf{Q}|\right]\Big|_{\mathbf{\Theta}_*} - \frac{1}{2}\ell(\mathbf{\Theta}_*)$$

$$\geq -\frac{1}{2}\left[\ell(\mathbf{\Theta}_*) + 1 - \sum_{i=1}^n \log\left(\lambda_i + \sigma^2\right)\Big|_{\mathbf{\Theta}_*}\right] \tag{47}$$

On the other hand, we have:

$$\mathbb{E}[g(\mathbf{x}_*; \mathbf{\Theta}_*)] = \mathbf{k}_*^\top \mathbf{Q}^{-1} \mathbf{y}$$

$$\leq \frac{1}{2}\left[\mathbf{k}_*^\top \mathbf{Q}^{-1}\mathbf{k}_* + \mathbf{y}^\top \mathbf{Q}^{-1}\mathbf{y}\right]\Big|_{\mathbf{\Theta}_*}$$

$$\leq \frac{1}{2}\left[\ell(\mathbf{\Theta}_*) + 1 - \sum_{i=1}^n \log\left(\lambda_i + \sigma^2\right)\Big|_{\mathbf{\Theta}_*}\right] \tag{48}$$

Thus, we have:

$$\mathbb{E}[g(\mathbf{x}_*; \mathbf{\Theta}_*)] = \pm\frac{1}{2}\left[\ell(\mathbf{\Theta}_*) + 1 - \sum_{i=1}^n \log\left(\lambda_i + \sigma^2\right)\Big|_{\mathbf{\Theta}_*}\right] \tag{49}$$

and by symmetry:

$$\mathbb{E}[g'(\mathbf{x}_*; \mathbf{\Theta}'_*)] = \pm\frac{1}{2}\left[\ell'(\mathbf{\Theta}'_*) + 1 - \sum_{i=1}^n \log\left(\lambda'_i + \sigma^2\right)\Big|_{\mathbf{\Theta}'_*}\right]$$

$$= \left(1 \pm \rho(\lambda, \sigma, \mathbf{\Theta}_*, \mathbf{\Theta}'_*)\right) \cdot \mathbb{E}[g(\mathbf{x}_*; \mathbf{\Theta}_*)] + \wp(\lambda, \sigma, \mathbf{\Theta}_*, \mathbf{\Theta}'_*) \tag{50}$$

where $\wp(\mathbf{\Theta}_*, \mathbf{\Theta}'_*)$ is a constant as defined below:

$$\rho(\lambda, \sigma, \mathbf{\Theta}_*, \mathbf{\Theta}'_*) \triangleq \max\left(\tau_{\lambda,\sigma}(\mathbf{K}), \tau_{\lambda,\sigma}(\mathbf{K}'), \frac{\lambda}{\sigma^2}\right)$$

$$\wp(\lambda, \sigma, \mathbf{\Theta}_*, \mathbf{\Theta}'_*) \triangleq \left(\sum_{i=1}^n \log\frac{\lambda_i + \sigma^2\big|_{\mathbf{\Theta}_*}}{\lambda'_i + \sigma^2\big|_{\mathbf{\Theta}'_*}}\right) \pm \rho(\lambda, \sigma, \mathbf{\Theta}_*, \mathbf{\Theta}'_*) \cdot \left(1 - \sum_{i=1}^n \log(\lambda_i + \sigma^2)\big|_{\mathbf{\Theta}_*}\right)$$

$\qquad\square$

## C  Model Parameterization and Practical Implementation

Our embedding algorithm is based on a VAE implementation where the latent prior, posterior and likelihood of the data generation process are represented via separate mixtures of $k$ Gaussian distributions over a $4$-dimensional space. For the latent prior, we set (and fixed) the means of each Gaussian component (i.e., the prior cluster means) at $k$ equidistant points on a $4$-dimensional sphere centered at zero with an optimizable radius. For the latent posterior and likelihood, the mean and covariance entries of each component in the mixture are parameterized as outputs of their respective neural networks, which we refer to as Gaussian nets.

In turn, the Gaussian nets are parameterized separately. Each starts with a linear layer comprising of $10$ neurons whose outputs are fed simultaneously to two separate hidden (linear) layers with $10$ hidden neurons each. Their outputs are then used to form the mean and covariance entries of the corresponding Gaussian component. All neurons are activated by a ReLU unit, and in addition, the (batch) outputs of the first linear layer are also standardized via a learnable 1D batch-norm layer to ensure the stability of batch optimization. The mixing weights that combine such Gaussian nets in the mixtures are also parameterized as the outputs of a linear layer with $k = 8$ neurons where $k = 8$ is also the number of components in our mixture.

The above parameterized latent prior, posterior, and likelihood are then connected in the variational lower-bound (ELBO) as expressed in the first two terms of Eq. (9). This ELBO objective is then combined with two regularization terms weighted with (manually tuned) parameters $\alpha = 8.0$ and $\beta = 1.2$ as detailed in Eq. (10). The entire function is optimized via gradient descent using the standard Adam optimizer with the default setting implemented in PyTorch.

Once learned, the outputs of the latent posterior were used as the encoded data which were fed as input to our revisited SSGP. For a practical implementation, we also found that additionally passing the encoded data to the latent likelihood generates a reconfigured version of the original data which helps to marginally improve the performance. All of our reported results below are generated with respect to this version of reconfiguration. All of our implementations of GP, SSGP and revisited SSGP that makes use of the output of this reconfiguration process, are also in PyTorch. Our experimental code is released at `https://github.com/hqminh/gp_sketch_nips`.

## D  Additional Empirical Results and Visualizations

This section provides additional empirical results and visualizations that complement and corroborate the reported results in the main text. In particular, we provide: (a) a more refined and comprehensive visualization of how our embedding algorithm (Section 3.3) re-configures data across different settings; and (b) an extended comparison with SSGP at different levels of sample complexity when evaluated on middle (10K data points) and large (500K data points) data sets. All data samples used in this section were extracted from the GAS SENSOR dataset [4][9].

### D.1  The Effect of Data Re-configuration: A Visual Demonstration

This section describes an ablation study to demonstrate the effectiveness of our data re-configuration component (i.e., to approximately meet the practical Conditions **1-3** of our refined analysis). Specifically, we demonstrate this by contrasting the scatter plots of data embeddings (see Fig. 3) before and after reconfiguration using our algorithm in Section 3.3 below. The visualizations are shown for 3 different samples of data, each of which has 10K data points.

For each data sample, its embedding was clustered and re-clustered before and after its reconfiguration. Both clustering processes were generated independently using K-Means to provide an objective visual measurement of the reconfiguration effects of our algorithm.

Observing the above visual excerpts, it appears that after reconfiguration, the clusters across different data samples all became significantly more disengtangled with a visibly increased distance between their cluster centers. This provides conclusive evidence to the data disengtangling effect of our

Figure 3: Visualizations of original (top) and reconfigured (bottom) data embeddings for 3 different (randomly selected) data samples annotated with S1, S2 and S3, respectively. Each visual excerpt is annotated with different colors corresponding to the different clusters that the data belong to. All visualizations are generated using T-SNE [41].

embedding algorithm. More importantly, this demonstration further reveals a practical aspect of data that has not been investigated before in the existing literature of GP:

**Data (especially experimental data) is often the manifestation of how latent concepts that underlie them were observed and depending on specific parameters of the observation process, these concepts might manifest differently in either more or less useful forms for learning. This raises the question of whether one can reorient the observation process to increase the utility of such data.**

In this vein of thought, to address the above question, our data reconfiguration algorithm can be considered to be one potential solution which uses a parameterized construction of a latent space to provide a handle on how to reorient the latent concepts that underlie our data. For an intuitive example, imagine how we would look at the outside world via a narrowed pigeonhole. With different viewing angles, we would perceive the same scene outside differently and apparently, some angles provide a much better perception of that scene (thus, allowing us to interpret the scene more accurately).

In technical terms, such a reorientation is implemented in our algorithm via the regularization of the mixture composition of the latent prior while constraining the entire embedding process to have it reflected on the latent posterior – see Eq. (10) – which was used to encode data into a latent space that exhibits the desired separation effect. Such separation/disentanglement is then shown (empirically) to be richer in information and can be leveraged to improve the sample complexity of SSGP (see Section D.2), thus supporting our theoretical analysis in Appendix A.

### D.2    Comparison with SSGP on Large Data

To demonstrate the effectiveness of the data disentanglement in reducing the sample complexity of SSGP, we compare the performance of SSGP and our revisited SSGP (which was instead applied on the reconfigured space of data) at different levels of sample complexity. All results were generated for two different data samples extracted from GAS-SENSOR [4]. One of these (containing 500K data points) is in fact on the same scale of the most extensive datasets used in the GP literature. All performance plots were visualized in Fig. 4 below. For each experiment, the data sample is divided into a train/test partition with an 8-2 ratio. All results were averaged over 5 independent runs.

We see that our revised SSGP consistently achieves better performance than its SSGP counterpart at all complexity levels. In particular, in all cases of the 10K setting, the performance of our revised

Figure 4: Graphs of performance comparisons between the full GP, our revised SSGP and the traditional SSGP on a $10K$ sample (a-c); $500K$ sample (d-f) and the entire GAS SENSOR dataset [4] totalling approximately 4M data points (g-i). In both settings, the performance differences were plotted at $p = 16$, $32$ and $64$. Note that for the $500K$ sized sample and the entire dataset (which contains 4M data points), the full GP model is not applicable due to its inability (memory- and computation-wise) to store and invert the corresponding large covariance matrix.

SSGP is also shown to approach closely that of the full GP, which serves as a gold-standard lower-bound on the achievable prediction error. This concludes our empirical demonstration which (we believe) has shown that with a proper reconfiguration of data, the predictive performance of a GP can be well-preserved at a much cheaper sample complexity as compared to the previous conservative estimate yielded by SSGP. In fact, the performance trend of SSGP as depicted in the above graphs shows that with more samples, it also slowly converges towards the performance level of GP and our revised SSGP but at a much greater sample complexity – see the shrinking performance gap between revisited SSGP and SSGP from Fig. 4d to Fig. 4e; and similarly, from Fig. 4g to Fig. 4h.