[Reviews · NeurIPS 2020]

Review 1

Summary and Contributions: This submission conducts sample complexity analysis for sparse spectrum Gaussian processes (SSGPs) when the input data exhibit certain clustering behavior. Further, a variational auto-encoder (VAE) based embedding algorithm is developed to project the raw input data on the hidden space where the desired regularity conditions are satisfied. Numerical tests validate that the proposed VAE-based SSGP exhibits improved performance relative to the conventional one.

Strengths: Commendable efforts to initiate sample complexity analysis for SSGPs under regularity conditions on the input data. Experiments demonstrate that the proposed SSGP method enjoys superior prediction performance relative to the original SSGP with the same number of spectral frequencies.

Weaknesses: 1. It is not clear how the VAE module couples with the SSGP for parameter training. It seems that the training objective for VAE (cf. Eq. (10)) relies on the parameters $\Theta$ of the GP. Further, is the dimension of the hidden variable $z$ the same as that of $x$? Further elaboration is due here. 2. Despite the claim that the cost of training the VAE is only linear in the number of data points, one would be interested to see the overall training time of the proposed SSGP relative to the plain SSGP. 3. In addition to RMSE-based evaluation, comparisons via the negative predictive log-likelihood that takes into account uncertainty would be recommended. 4. The results in this submission seem to hold for Gaussian kernels only. How about other shift-invariant GP kernels? A remark is expected in this aspect. 5. The result of sample complexity is only mentioned in the text after Theorem 2. Since this is a major result, it is recommended to state it in Theorem 2. 

Correctness: The claims and method seem correct. The proofs in the supplementary file have not been checked thoroughly.

Clarity: This paper is well written in general, but clarity could be improved. Please refer to the comments earlier.

Relation to Prior Work: Yes, differences from the most related works [3, 27, 28] have been discussed.

Reproducibility: Yes

Additional Feedback: ==update after rebuttal== The rebuttal letter has addressed most of my concerns regarding the NLL performance and training time. What is due for clarification is the explicit form of the VAE training objective; e.g., how about the additional term in (9), as mentioned also by Rev 4. Based on these considerations, I will increase my score from 6 to 7.


Review 2

Summary and Contributions: The paper proposes a novel approximation to sparse-spectrum Gaussian processes. The authors also provide a data partitioning algorithm with better sample complexity.

Strengths: The theoretical results seem sound. The experimental results are encouraging.

Weaknesses: Condition 1 seems very restrictive. (Please see my additional feedback below.)

Correctness: The main claims seem correct.

Clarity: The paper is clear.

Relation to Prior Work: Relation to prior work is partially addressed. (Please see my additional feedback below.)

Reproducibility: Yes

Additional Feedback: Some literature in the estimation of Gram matrices needs to be discussed: - Rahimi and Recht, "Random features for large-scale kernel machines", NeurIPS 2007. - Sriperumbudur and Szabo, "Optimal rates for random Fourier features", NeurIPS 2015. - Sutherland and Schneider, "On the error of random Fourier features", UAI 2015. Please discuss why Condition 1 is reasonable and not too restrictive. More specifically, please discuss the specific rates of the number of Gaussian components in the mixture b \in O(log n), the mixture weights \pi_i \approx 2^{i/2} as well as the variances \gamma_i \in O(1/\sqrt{d}). === AFTER REBUTTAL I am not too satisfied with the tangential answer given by the authors regarding Condition 1. I was specifically asking about the particular rates of b, \pi_i and \gamma_i. Given the above, I keep my initial evaluation of 6.


Review 3

Summary and Contributions: The paper exploits the clustering structure of the data to provide an approximation for the Gaussian Processes using mixtures of Gaussians, and develops a related algorithm based on finding the clustering of the data in a latent space using the variational auto-encoders techniques.

Strengths: Well-written, clear analysis, and interesting clustering algorithm

Weaknesses: It is unclear how frequently Conditions 1-3 are satisfied in practice

Correctness: Yes

Clarity: Yes

Relation to Prior Work: Yes

Reproducibility: Yes

Additional Feedback: The main question that I have that does not seem to be discussed is the complexity of the training step in the proposed scheme. Presented results were motivated by a high computational cost of learning and inference O(n^3) for the general Gaussian Processes; is the computational overhead introduced by VAEs to discover the latent space where the data may be clustered worth it for large n? What is the computational cost introduced? Is it possible that the resulting scheme will be even more expensive compared to the full GP procedure? ------------- Post-rebuttal: I am satisfied with the additional clarifications that the authors produced in their rebuttal. It would be beneficial to include these considerations in the final version of the paper. I think that the paper is a good study, while I still have some doubts about practicality meeting the conditions theoretically studied in the paper (no guarantees for a larger latent space); hence overall I don't see any particular reason to change my score, so I leave it as is (7 - accept).


Review 4

Summary and Contributions: The paper revisits the bounds for sparse gaussian processes, in the case where the data is distributed as a mixture of gaussian clusters. This theoretic analysis comes with very encouraging bounds that suggest a promising avenue for research. the paper also comes with the idea of training an auto-encoder that allows going from the input space to a latent space with the appropriate clustered structure.

Strengths: * The paper is very clear on both GP and prior work for sparse GP approximation. It flows naturally and provides adequate references * the motivations are quite clear * The contributions are at a theoretical level and do leave space for fruitful research. Basically, the authors show that bounds for GP inference on clustered data can be made super tight and computationally interesting. The idea to go from input space to a latent space where this clustered structure is preserved is interesting. * I appreciated the quality of the available github implementation

Weaknesses: * the experimental section is rather weak. Although results are shown as interesting, scalability is not demonstrated. Still, I would say that the contribution is rather at a theoretical level and that these experiments can be considered sufficient * I think there is a point where the paper somehow looses its initial rigour. This happens where the VAE scheme is introduced. it's ok to me, but I would suggest the authors acknowledge that that part is a baseline attempt at producing a clustered feature space. Indeed, it would make it easier for the reader to understand where room for improvement stands.

Correctness: I couldn't check everything, but the paper definitely looks correct to me.

Clarity: yes

Relation to Prior Work: A reference to the reparameterization trick could be beneficial

Reproducibility: Yes

Additional Feedback: • P2L68: although it's super common, it does not read obvious to me at first read that your further analysis is limited to this particular case of a gaussian (squared exponential SE) kernel • Related to the question above: although the SE kernel is common, do you think that the approach could be applicable to some that are more exotic ? For instance by seing them as sums of SE ? or some other trick ? • (9): how is the additional term computed in practice ? I can see the code is nicely available, it would be good to quickly advertise it here to reassure the anxious reader ? • I would say that the experimental section is a bit disappointing. The method is advertised for scaling up, but experiments report a few thousand data points only. I would have appreciated someting in the vein of millions. • P6L247-onwards: it looks to me that the choice of an auto-encoder is arguable. In a sense, I see this as a very conservative and pessimistic way to go for this clustering. Indeed, as far as I understand, the decoder is actually not used, but only as a regularizer ensuring that the trained clusters are indeed capturing enough to reconstruct the training data. It is actually likely that much less than this is required: maybe the GP to train only needs some super specific part of information, that would actually not need full reconstruction. put it otherwise: I wonder whether some joint training of the GP and the parameters would be feasible, with some cost function for the decoder that would be weaker than reconstruction. --- after rebuttal, I notice that the authors decided not to take my comments into account. lowered score in accordance: 1. they only consider toy datasets (thousands datapoints) everywhere , while.claiming.scalability 2. nothing IMO justifies the use of an auto-encoding criterion for the clustering procedure, and it loo's to me, again, like a conservative/pessimistic approach. This is not even discussed.

[Author Response · NeurIPS 2020]

**To all reviewers:** We thank all the reviewers for providing valuable feedback. We have included additional experiments
(on the GAS dataset with 10k data points) below to compare the scalability and performance of Revisited SSGP (rSSGP)
with $p = 32$ samples to that of SSGPs with $p = 32$ and $p = 1024$ samples. Fig. 1(a) shows that rSSGP outperforms
SSGP in terms of NLL of prediction (and RMSE, which is omitted here due to space constraints and will be included in
the revised version) even when SSGP is using many more samples. Fig. 1(b) shows the runtime tradeoff incurred by
the overhead of VAE training. While rSSGP expectedly runs slower than SSGP with the same number of samples, it
outperforms and runs faster than SSGP with 1024 samples, which implies that rSSGP is more scalable than SSGP with
the same performance (i.e., $p > 1024$). Fig. 1(c) shows the distribution of embedded data fitted to a Gaussian mixture
model with 8 components (the 2D projection onto the span of the first two eigenvectors). The latent inputs form clusters
with varying density (we plot this for convenience because each cluster covariance is an $18 \times 18$ matrix). The mixture
weights are $[0.046, 0.063, 0.070, 0.130, 0.153, 0.155, 0.167, 0.215]$ which roughly corresponds to the $2^{-i}$ values as
required by our practical conditions. As a consequence, we observe $85\%$ of latent input pairs to be cross-cluster pairs,
which expectedly correspond to small kernel entries in our analysis.

(a) NLL of prediction    (b) Training time    (c) Latent input distribution

Figure 1: Comparative performance of Revisited SSGP ($p = 32$) and SSGP ($p = 32$ and $1024$) on GAS (10k) dataset.

**R1: VAE training:** Both VAE's reconstruction loss and SSGP's NLL are combined additively into one loss function,
which enables end-to-end training of both VAE's and SSGP's parameters. The dimensions of $x$ and $z$ are not necessarily
the same (e.g., $|x| = 8$ and $|z| = 4$). Compressing $x$ into $z$ (with a Gaussian mixture prior on the VAE) allows us to
reconfigure the data onto a latent space conditioned to exhibit the disentanglement that enables our theoretical analysis
(please refer to Appendix D1 and our visualization above). **Analysis for other kernels:** Extending our current analysis
to other kernels is feasible since the spectral theorem applies generally to many shift-invariant GP kernels [1, 2]. In fact,
we can modify Lemmas 3 and 4 in Appendix A (Equations 13-17) to make our analysis applicable to a broader range of
exponential kernels. We will include this. **Sample complexity result:** We will state it in Theorem 2 as you suggested.

**R2: Suggested literature:** We have discussed (Rahimi and Recht) in lines 96-104 and in footnote 3. We will state this
explicitly in the introduction and include the other works in our discussion. While these works generate bounds on SSGP
that only hold for a fixed parameter configuration, ours hold universally on the entire parameter space. **Restrictiveness
of practical conditions:** Our analysis will hold (approximately) if we can reconfigure and embed the data onto a latent
space that exhibits such conditions. As such, in 3.3.2, we adopt a VAE with a Gaussian Mixture prior to embed the raw
data (please see Appendix D1 and the extra plots above) such that our conditions are likely to hold.

**R3: Practicality of conditions:** Please see the above response for R2. **Training complexity**: The VAE's complexity
per update iteration is $\mathcal{O}(b \cdot \text{poly}(p))$ where $p$ is the number of VAE parameters and $b$ is the batch size. For large
datasets, $n \gg p$ so the overhead is very mild with respect to $n$. To the best of our knowledge, both VAE and GP have
no guarantee for convergence in terms of total no. iterations. We show empirically (see above plot) that the resulting
scheme will not be slower than normal SSGP with sufficiently many spectral samples to produce similar performance
and surely faster than full GP with a per-iteration update cost of $\mathcal{O}(n^3)$. In addition, for full GP prediction, the $\mathcal{O}(n^2)$
memory cost is a bottleneck, whereas our scheme allows batch training and does not incur this expensive memory cost.

**R4:** Thank you for recognizing our theoretical contribution. **Rigor of VAE:** The proposed VAE is a practical measure
to achieve the conditions that enable our analysis and is more than a baseline attempt to cluster data. We will put a
remark to clarify as suggested. **Analysis for other kernels:** Please refer to our response for R1. **Computation of the
additional term:** The model is optimized via updating its parameters along the direction of the gradient. While the
exact gradient is not tractable, its unbiased stochastic estimate can be computed using the reparameterization trick as
described in [22] (a standard practice in many VAE works). **Large-scale experiments:** In Appendix D2, we show
results on a large dataset with 0.5 million data points, on which full GP is already infeasible. We will include extra
results to showcase our performance. **Unused decoder:** While the reconstructed data is not used for prediction, it is
useful as an auxiliary training objective to obtain our practical conditions. It would be ideal if there is an alternative
approach to ensure this happens in a more direct manner and reduce the difficulty of the learning problem, which would
in turn allow us to tighten the sample complexity further. This is an interesting direction to take for future research.

[1]: Fourier Feature Approximations for Periodic Kernels in Time-Series Modelling (Tompkins and Ramos, 2018).
[2]: Generalized Spectral Kernels (Samo and Roberts, 2015).

[Meta-Review · NeurIPS 2020]

This paper performs a new analysis of the sample complexity of Gaussian processes when the Gaussian covariance is approximated by sparse random Fourier features. They show that if one assumes that the data are clustered, then the approximate covariance matrix is close to the true covariance matrix, and in turn that the approximate GP estimator is close to the GP estimator, for a number of sparse features smaller than what needs without the clustering assumption. In a second part, the paper proposes to combine GP with a prior embedding of data to a latent space where they are clustered, using a VAE, and report preliminary experimental results. All reviewers found the theoretical work interesting, novel, and worth presenting at NeurIPS. Exploiting a cluster structure in the data to decrease the number of random features needed to approximate the kernel is a nice idea, and this paper provides a rigorous analysis to quantify how this happens. Reviewers were a bit more critical about the second part, where it is not very clear why the objective function of the VAE is a good one, and had questions about the computational cost of the overall procedure (given that the goal of using a sparse spectrum approximation is to gain in computational cost). They also did not find the experiments very impressive, and suggested in particular to consider larger datasets to assess the scalability of the approach.